# CCN1 is an opsonin for bacterial clearance and a direct activator of Toll-like receptor signaling

Joon-Il Jun [1] & Lester F. Lau [1✉]

Expression of the matricellular protein CCN1 (CYR61) is associated with inflammation and is required for successful wound repair. Here, we show that CCN1 binds bacterial pathogen-associated molecular patterns including peptidoglycans of Gram-positive bacteria and lipopolysaccharides of Gram-negative bacteria. CCN1 opsonizes methicillin-resistant *Staphylococcus aureus* (MRSA) and *Pseudomonas aeruginosa* and accelerates their removal by phagocytosis and increased production of bactericidal reactive oxygen species in macrophages through the engagement of integrin $\alpha_v\beta_3$. Mice with myeloid-specific *Ccn1* deletion and knock-in mice expressing CCN1 unable to bind $\alpha_v\beta_3$ are more susceptible to infection by *S. aureus* or *P. aeruginosa*, resulting in increased mortality and organ colonization. Furthermore, CCN1 binds directly to TLR2 and TLR4 to activate MyD88-dependent signaling, cytokine expression and neutrophil mobilization. CCN1 is therefore a pattern recognition receptor that opsonizes bacteria for clearance and functions as a damage-associated molecular pattern to activate inflammatory responses, activities that contribute to wound healing and tissue repair.

[1] Department of Biochemistry and Molecular Genetics, College of Medicine, The University of Illinois at Chicago, 900 South Ashland Avenue, Chicago, IL 60607, USA. ✉email: lflau@uic.edu

The rapid emergence of antibiotic-resistant bacteria poses a serious threat to public health globally[1]. Despite the remarkable success of antibiotics in treating bacterial infections since the 1940s, resistance to virtually all available antibiotics has been observed in recent decades. Antibiotic-resistant bacterial infections account for nearly 3 million clinical cases and >35,000 deaths annually in the United States alone[2], and >700,000 deaths worldwide[3]. Among the most encountered pathogens are the Gram-positive *Staphylococcus aureus* (*S. aureus*) and the Gram-negative *Pseudomonas aeruginosa* (*P. aeruginosa*). These bacteria quickly acquire antibiotic resistance, largely through mobile genetic elements such as plasmids and transposons[4]. Methicillin-resistant *S. aureus* (MRSA) alone accounts for a large portion of nosocomial[5] and community-acquired infections[6] and is a leading cause of endocarditis, septicemia, and skin and soft-tissue infections (SSTIs)[7,8]. *P. aeruginosa* is an opportunistic pathogen commonly found in ventilator-associated pneumonia, catheter-associated urinary tract infections, burn wounds, and blood infections[9]. Although vaccines are under development for these pathogens, none have achieved clinical efficacy to date[10,11]. Therefore, there is an urgent need to identify effective therapeutic strategies for such antibiotic-resistant pathogens as *S. aureus* and *P. aeruginosa*.

Upon bacterial infection, vertebrate hosts mount an onslaught of attacks at the invaders, including activation of the complement system that targets bacteria for lysis or phagocytosis. In addition, resident macrophages that patrol the tissues eliminate microbes by phagocytosis, whereupon the activated macrophages produce cytokines and chemokines to induce inflammation and to recruit immune cells that reinforce bacterial removal and initiate adaptive immunity[12]. Accordingly, defects in phagocytes lead to impaired host defense against infections[13]. Bacterial pathogens are typically detected by host proteins known as pattern recognition receptors (PRRs) that recognize and bind distinct microbial components, or pathogen-associated molecular patterns (PAMPs)[14]. Among the best characterized PRRs are cell surface Toll-like receptors (TLRs), which are potent activators of inflammatory responses upon recognition of PAMPs[15,16]. TLRs can also mediate sterile inflammation by interacting with damage-associated molecular patterns (DAMPs), endogenous danger molecules that are released from damaged or dying cells[17,18]. Professional phagocytes express PRRs that bind PAMPs and induce phagocytosis directly, such as macrophage mannose receptor and macrophage scavenger receptors[19,20], as well as opsonins that recognize PAMPs and connect the bacteria to phagocytes for elimination[21,22]. As these mechanisms of host defenses are unrelated to the bactericidal pathways of antibiotics, treatments that stimulate or enhance these host defenses may provide therapeutic strategies for the clearance of microbial pathogens irrespective of their antibiotic resistance status.

*Ccn1* (*Cyr61*) encodes a 40 kDa matricellular protein that is essential for embryonic development[23,24]. In adults, *Ccn1* expression is associated with inflammation and is required for successful injury repair in the skin[25,26], liver[27], and gut[28]. Structurally, CCN1 is organized into four conserved domains with homologies to insulin-like growth factor-binding protein (IGFBP), von Willebrand factor type C repeat (vWC), thrombospondin type-1 repeat (TSR), and a cysteine-knot motif in the C-terminal (CT) domain (Supplementary Fig. 1)[29]. Mechanistically, CCN1 acts through direct binding to specific integrin receptors in a cell type-specific manner, engaging coreceptors in some contexts[30,31]. The specific CCN1-binding sites for several integrin receptors, including $\alpha_v\beta_3/\alpha_v\beta_5$, $\alpha_6\beta_1$, and $\alpha_M\beta_2$, have been identified[29,30].

We have recently shown that CCN1 promotes efferocytosis, or phagocytosis of apoptotic cells, by binding phosphatidylserine,

the "eat-me" signal on apoptotic cells and bridging them to macrophages for elimination through engagement of the phagocytic receptor, integrin $\alpha_v\beta_3$[25]. Consequently, CCN1 stimulates the removal of apoptotic neutrophils and accelerates wound healing progression. Here, we have found that CCN1 functions as an opsonin for bacterial clearance through specific binding to PAMPs of *S. aureus* and *P. aeruginosa*, inducing their phagocytic elimination through engagement of integrin $\alpha_v\beta_3$ on phagocytes. Additionally, CCN1 enhances bacterial killing in macrophages by increasing the production of bactericidal reactive oxygen species (ROS). Independent of its opsonin activity, CCN1 can also directly bind and activate TLR2 and TLR4 to induce the expression of proinflammatory cytokines in a MyD88-dependent manner. Altogether, these results show that CCN1 is a PRR that induces opsonophagocytosis of bacteria and a DAMP that can trigger sterile inflammation through activating TLR signaling.

## Results

**CCN1 binds *S. aureus* and *P. aeruginosa* through PAMPs.** Based on the finding that CCN1 induces efferocytosis of apoptotic neutrophils[25], we investigated the possibility that CCN1 may also target bacterial pathogens for removal by phagocytosis, an activity that requires specific recognition of bacterial PAMPs. Thus, we first assessed whether CCN1 can recognize and bind the Gram-positive *S. aureus* and the Gram-negative *P. aeruginosa* in a solid-phase-binding assay. Indeed, *S. aureus* showed efficient binding to immobilized CCN1, with half-maximal binding occurring at ~10 pmol per well CCN1 (Fig. 1a). Soluble CCN1 also bound *S. aureus* efficiently as observed by flow cytometry (Supplementary Fig. 2). *S. aureus* can bind various host extracellular matrix (ECM) proteins by expressing bacterial adhesins, including the fibronectin-binding protein (FnBP)[32], although the specific *S. aureus* strain (MRSA USA300) used in this study does not express the collagen adhesin (Cna)[33]. Consistently, *S. aureus* bound immobilized fibronectin (FN) in a dose-dependent manner with half-maximal binding at ~20 pmol per well but failed to bind collagen I (Col1α1; Fig. 1a). Both the CCN1-D125A mutant protein[34], which is unable to bind integrins $\alpha_v\beta_3/\alpha_v\beta_5$ as a result of a single amino acid substitution (Asp125 to Ala), and the CCN1-DM mutant[35], which is defective for binding integrins $\alpha_6\beta_1/\alpha_M\beta_2$, were able to bind *S. aureus* with affinities similar to that of wild type (WT) CCN1 (Fig. 1b). Thus, the CCN1-binding site(s) for *S. aureus* are distinct from those for integrins $\alpha_v\beta_3/\alpha_v\beta_5$ and $\alpha_6\beta_1/\alpha_M\beta_2$.

We endeavored to identify the region of CCN1 responsible for binding *S. aureus* using a series of deletion mutants (Supplementary Fig. 1). Mutants that contain the TSR domain (ΔCT and TSR alone) showed as strong binding to *S. aureus* as CCN1-WT, whereas mutants containing the vWC domain (IGFBP-vWC and vWC alone) displayed strong to moderate binding (Fig. 1c). However, the IGFBP domain alone did not bind *S. aureus*. These results suggest that CCN1 binds *S. aureus* through the TSR and vWC domains, whereas the CT domain is dispensable (Fig. 1c). Prominent among PAMPs of Gram-positive bacteria are the cell wall components peptidoglycan (PGN) and lipoteichoic acid (LTA). We found that CCN1 proteins (WT, D125A, or DM) bound to immobilized PGN, but not LTA (Fig. 1g). Immunoblotting also showed that both the vWC and TSR domains could bind PGN individually, but not the IGFBP domain (Supplementary Fig. 3). These results demonstrate that CCN1 directly binds *S. aureus* through its vWC and TSR domains by recognizing PGN, a bacterial PAMP.

CCN1 also exhibited strong binding to *P. aeruginosa* in a dose-dependent manner, with half-maximal binding at ~20 pmol per well CCN1 (Fig. 1d). Other ECM proteins, including FN, LN, and

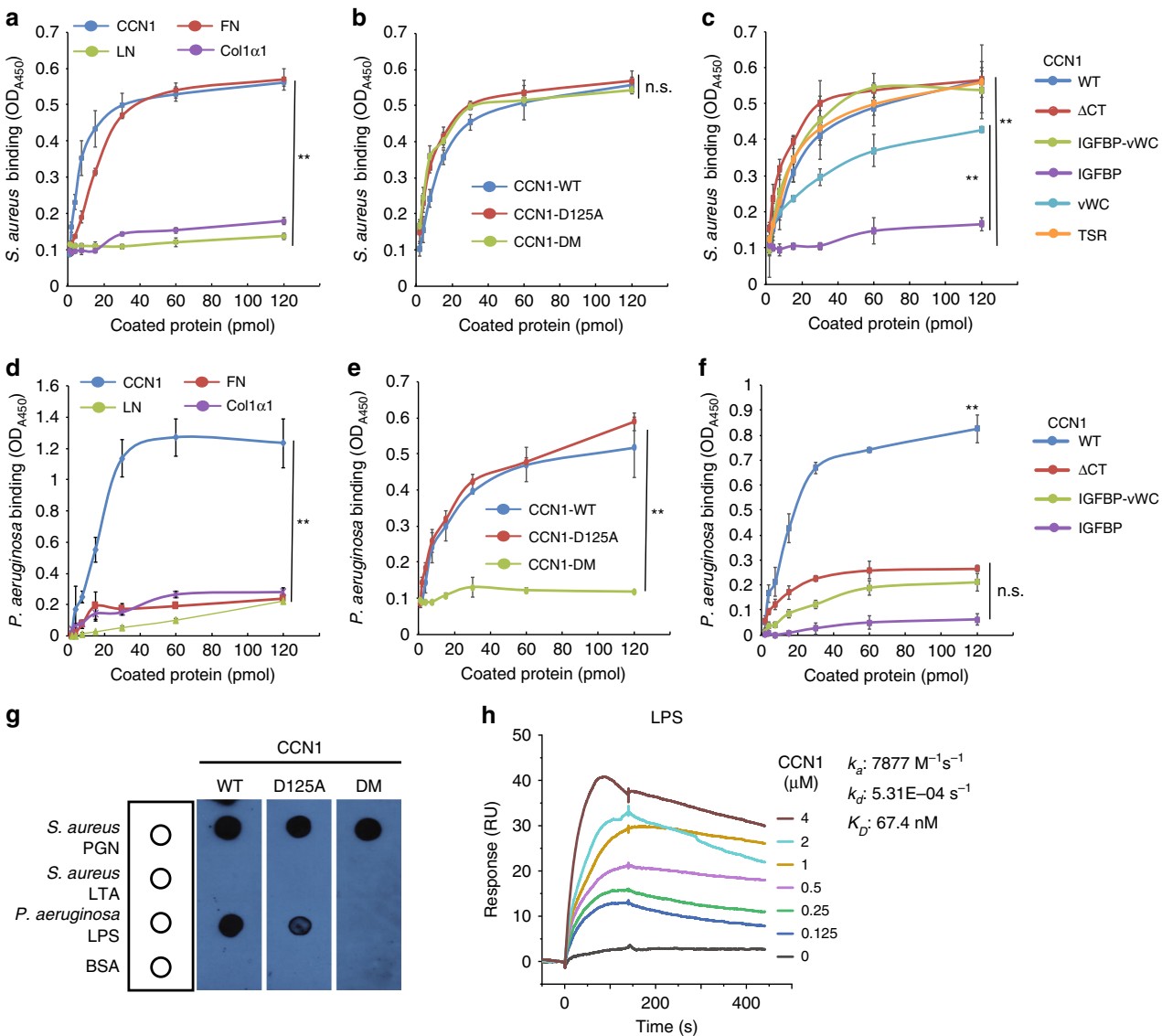

**Fig. 1 CCN1 binds *S. aureus* and *P. aeruginosa* through PGN and LPS, respectively.** CCN1 binding to *S. aureus* (**a–c**) and *P. aeruginosa* (**d–f**) was evaluated using a solid-phase-binding assay. **a** *S. aureus* bound to surfaces coated with CCN1-WT or other ECM proteins (FN fibronectin, LN laminin, Col1α1 collagen type I) was quantified using polyclonal anti-*S. aureus* antibodies. CCN1 and FN bound to *S. aureus*, whereas LN and Col1α1did not. **b** Binding assay of *S. aureus* to recombinant CCN1-WT, CCN1-D125A, and CCN1-DM proteins as above. **c** Binding of *S. aureus* to CCN1 deletion mutant proteins as above. **d** Solid-phase-binding assays for *P. aeruginosa* were performed as above and detected with polyclonal anti-*P. aeruginosa* antibodies. **e** Binding assay showing *P. aeruginosa* binding to CCN1-WT and CCN1-D125A, but not CCN1-DM. **f** Solid-phase-binding assays with CCN1 deletion mutants. All assays were done in triplicates and data are expressed as mean ± s.d. Statistical evaluation was performed by one-sided, two-sample with equal variance *t*-tests. \*\**p* < 0.01, n.s. = not significant. **g** Bacterial molecular patterns (PGN of *S. aureus*; LTA of *S. aureus*; LPS of *P. aeruginosa*; 1 μg each) and BSA as control spotted on nitrocellulose membranes were incubated with CCN1 proteins (WT, D125A, or DM; 2 μg each in PBS) and bound CCN1 was visualized using anti-CCN1 antibodies. Dot blot shown is a representative image. **h** Senograms of SPR analysis of CCN1 binding to LPS. CCN1 at various concentrations was used as analyte to detect binding to LPS immobilized on HPA hydrophobic sensor chip. $K_a$, $K_d$, and $K_D$ values were obtained by kinetic analysis.

Col1α1, did not show binding (Fig. 1d). Remarkably, whereas CCN1-WT and CCN1-D125A bound *P. aeruginosa* with similar dose-dependence, CCN1-DM was completely unable to bind, indicating that the essential binding site for *P. aeruginosa* was disrupted in CCN1-DM (Fig. 1e). Consistently, all deletion mutants lacking the CT domain tested (ΔCT, IGFBP-vWC, IGFBP) failed to bind *P. aeruginosa* (Fig. 1f). Mutations in CCN1-DM changed several lysine residues and an arginine to glycines in the CT domain (Supplementary Fig. 1), rendering CCN1 unable to bind heparin, a polysaccharide of the glycosaminoglycan family[35,36]. Thus, we postulated that CCN1-DM may also be defective for binding lipopolysaccharides (LPS), a PAMP of

Gram-negative bacteria. Indeed, both CCN1-WT and CCN1-D125A bound immobilized LPS from *P. aeruginosa*, but not CCN1-DM (Fig. 1g).

We further performed surface plasmon resonance (SPR) analysis to evaluate the binding of CCN1 to LPS, which was immobilized on an HPA hydrophobic sensor chip (Fig. 1h). Sensorgrams showed high-affinity CCN1 binding to LPS with a $K_D$ of 67.4 nM. CCN1 also bound to the Gram-positive *Streptococcus pneumoniae* and the Gram-negative *Salmonella typhimurium*, supporting the notion that CCN1 may bind a broad spectrum of bacterial species through PAMPs (Supplementary Fig. 4). These results show that CCN1 is a PRR that can

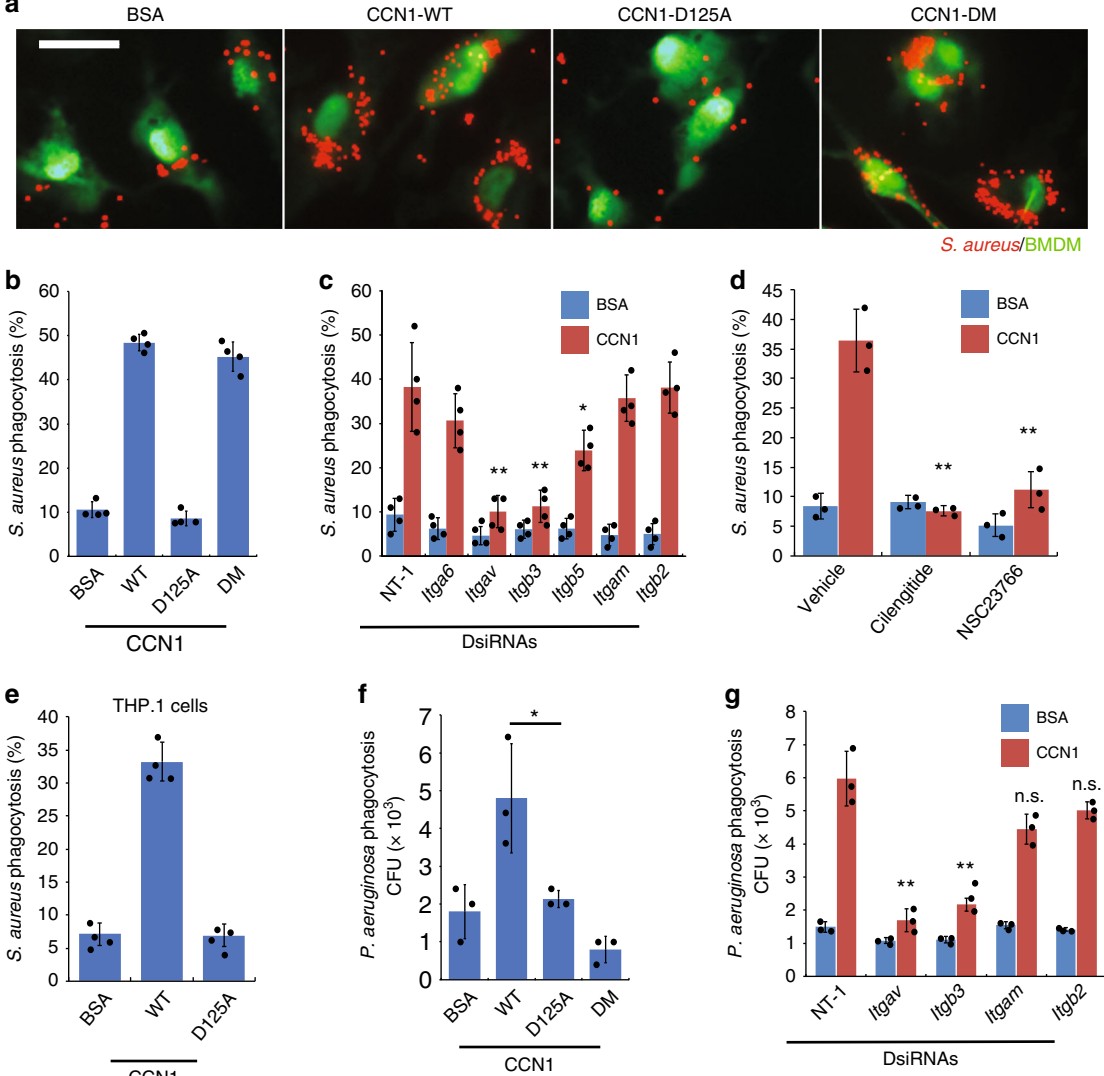

**Fig. 2 CCN1 stimulates phagocytosis of bacteria through $\alpha_v$ integrins in macrophages.** BMDMs (labeled with CellTracker™ Green) were pretreated with either BSA, CCN1-WT, CCN1-D125A, or CCN1-DM proteins (2 μg per ml) followed by incubation with bacteria for 45 min. **a** Phagocytosis of *S. aureus* (pHrodo Red conjugates) is shown in representative images from fluorescence microscopy. Scale bar = 40 μm. **b** Quantification of phagocytosis of *S. aureus* by BMDMs pretreated with various proteins. **c** BMDMs were transfected with DsiRNAs to knockdown integrins $\alpha_6$, $\alpha_v$, $\beta_3$, $\beta_5$, $\alpha_M$, or $\beta_2$; knockdown efficiencies are shown in Supplementary Fig. 6. Cells were then pretreated with CCN1 or BSA, and phagocytosis of *S. aureus* was quantified. NT-1, non-targeting RNA. **d** Effect of cilengitide (1 μM) and the Rac1 inhibitor NSC23766 (10 μM) on CCN1-induced phagocytosis of *S. aureus*. **e** CCN1-induced phagocytosis in differentiated THP.1 human macrophages. **f** BMDMs were pretreated as above and then incubated with live *P. aeruginosa* (MOI of 10) for 45 min, followed by gentamycin (200 μg per ml for 15 min) treatment. Phagocytosis was measured by enumeration of bacterial colonies grown from cell lysates. **g** Effects of DsiRNA-mediated knockdown of integrins $\alpha_v$, $\beta_3$, $\alpha_M$, or $\beta_2$ on CCN1-induced phagocytosis of *P. aeruginosa*. All data are expressed as mean ± s.d. in triplicate determinations. Statistical evaluation was performed by one-sided, two-sample with equal variance *t*-tests. *$p < 0.05$, **$p < 0.01$, n.s. = not significant. Source data are provided as a Source Data file.

directly recognize and bind bacterial PAMPs, including PGN in Gram-positive bacteria and LPS in Gram-negative bacteria.

**CCN1 is an opsonin that stimulates phagocytosis of bacteria.** Given the ability of CCN1 to bind bacterial PAMPs (Fig. 1) and to stimulate phagocytosis[25], we hypothesized that CCN1 may act as an opsonin to stimulate bacterial clearance by phagocytosis. To test this possibility, we incubated bone marrow-derived macrophages (BMDMs; labeled green) with *S. aureus* bioparticles (pHrodo® conjugates, labeled red). Engulfment of *S. aureus* by macrophages was observed by fluorescence microscopy within 45 min (Fig. 2a). The addition of CCN1 protein stimulated

engulfment by ~5-fold (Fig. 2a, b). Inhibition of actin polymerization with cytochalasin D abrogated bacterial uptake, confirming phagocytosis (Supplementary Fig. 5). The CCN1-D125A mutant protein failed to enhance phagocytosis, whereas CCN1-DM was fully active (Fig. 2a, b), indicating that $\alpha_v$ integrins are the essential receptors for CCN1-induced phagocytosis. Consistently, dicer-substrate short interfering RNAs (DsiRNA)-mediated knockdown of either integrin $\alpha_v$ (*Itgav*) or $\beta_3$ (*Itgb3*) efficiently blocked CCN1-induced phagocytosis, whereas knockdown of $\alpha_6$ (*Itga6*), $\alpha_M$ (*Itgam*), or $\beta_2$ (*Itgb2*) did not (Fig. 2c and Supplementary Fig. 6). Knockdown of integrin $\beta_5$ (*Itgb5*) only partially inhibited phagocytosis, suggesting that integrin $\alpha_v\beta_3$ is the primary phagocytic receptor, whereas $\alpha_v\beta_5$ plays an auxiliary

role. Thus, cilengitide, a cyclic-RGD peptide that selectively blocks integrins $\alpha_v\beta_3/\alpha_v\beta_5$, obliterated CCN1-induced phagocytosis (Fig. 2d). Likewise, CCN1-WT, but not CCN1-D125A, enhanced phagocytic uptake of *S. aureus* in bone marrow-derived neutrophils (BM-PMNs; Supplementary Fig. 7) and in differentiated THP.1 human macrophages (Fig. 2e), indicating the involvement of the same receptors in CCN1-induced phagocytosis of bacteria in neutrophils and human macrophages. Of note, integrin $\alpha_M\beta_2$ is also known as CR3 (CD11b/CD18), the type 3 complement receptor that binds iC3b to promote phagocytosis[37], but is not involved in CCN1-induced phagocytosis based on these mutational (CCN1-DM) and DsiRNA knockdown results. Consistent with previous findings that $\alpha_v\beta_3$-mediated phagocytosis requires activation of Rac1[25,38], inhibition of Rac1 by NSC23766 blocked CCN1-induced phagocytosis (Fig. 2d). Concordant with the role of PI3K in Rac1-dependent cytoskeletal rearrangement[39], PI3K inhibitor LY294002 reduced CCN1-induced phagocytosis (~50%), whereas ERK inhibitor (PD98059) or p38 MAPK inhibitor (SB203580) had no effect (Supplementary Fig. 5). Furthermore, commercially available CCN1 protein from different cellular sources also bound PGN and LPS and induced cilengitide-inhibitable phagocytosis of *S. aureus* (Supplementary Fig. 8), suggesting that these reproducible functions are bona fide activities of CCN1.

To evaluate whether CCN1 can also induce phagocytosis of Gram-negative bacteria, we tested its effects on *P. aeruginosa* (Fig. 2f, g). BMDMs were incubated with *P. aeruginosa* and phagocytosed bacteria were enumerated after extracellular bacteria were eliminated by gentamycin. We found that CCN1 also enhanced phagocytic uptake of *P. aeruginosa*, whereas CCN1-D125A was inactive (Fig. 2f). CCN1-DM, which was fully able to stimulate phagocytosis of *S. aureus*, was completely unable to enhance phagocytosis of *P. aeruginosa* (Fig. 2f). That CCN1-DM was incapable of binding LPS or *P. aeruginosa* (Fig. 1e, g) provided a clear explanation for this finding and indicated that binding to bacteria through PAMP is critical for CCN1-induced bacterial clearance. Knockdown of either $\alpha_v$ or $\beta_3$ blocked CCN1-induced phagocytosis of *P. aeruginosa*, whereas knockdown of $\alpha_M$ or $\beta_2$ had little effect (Fig. 1g). These results show that CCN1 opsonizes both Gram-positive and Gram-negative bacteria for phagocytosis, engaging integrin $\alpha_v\beta_3$ as the primary phagocytic receptor.

**CCN1 increases ROS for bacterial killing after phagocytosis.** Upon phagocytosis, ingested bacteria are trapped in vacuoles, or phagosomes, which undergo maturation to become phagolysosomes with diverse antimicrobial arsenals, including microbicidal peptides, lipases, and capacity to generate reactive oxygen species (ROS)[40]. CCN1 is known to induce ROS production in other biological contexts[26,41], prompting us to test whether CCN1 can facilitate bacterial killing following phagocytosis by inducing ROS. To monitor bacterial killing, we allowed BMDM phagocytosis of *S. aureus* to proceed for 90 min and eliminated the remaining extracellular bacteria by lysostaphin[42]. BMDMs were then treated with CCN1 and the number of surviving intracellular bacteria was monitored over time. As shown in Fig. 3a, the viable *S. aureus* count in CCN1-treated BMDMs was significantly lower than that of bovine serum albumin (BSA)-treated control after a 3 h chase period, indicating that CCN1 accelerated bacterial killing after phagocytosis.

Next, we examined whether CCN1 stimulated ROS production to enhance bacterial killing. Treatment of BMDMs with CCN1-induced ROS production within 30 min as judged by staining with dihydroethidium (DHE), a superoxide ($O_2^-$) detection dye (Fig. 3b, c). CCN1-DM also induced ROS production but not

CCN1-D125A, indicating that CCN1 acts through $\alpha_v$ integrins (Fig. 3b, c). Accordingly, cilengitide effectively abolished CCN1-induced ROS and subsequent bacterial killing (Fig. 3d, e). In macrophages, NADPH oxidase 2 (NOX2) is the principal enzyme responsible for superoxide production[43]. Whereas Rac1 is an activating subunit of NOX2 in macrophages, Rac2 activates NOX2 in neutrophils[44,45]. Consistently, the Rac1 inhibitor NSC23766 blocked CCN1-induced ROS production and bacterial killing (Fig. 3f, g). Apocynin, a non-isoform-specific NOX inhibitor and a ROS scavenger, exhibited strong inhibitory effects (Fig. 3f, g). The NOX2 inhibitor (GSK2795039), but not NOX1 inhibitor (ML-171), diminished CCN1-induced ROS production and bacterial killing (Fig. 3f, g). These results show that CCN1 enhances microbial killing after phagocytosis by the induction of ROS through integrin $\alpha_v$-mediated activation of Rac1-NOX2.

**CCN1 is critical for bacterial clearance in animal models.** Given that CCN1 opsonizes bacteria for phagocytosis and killing, we investigated the role of CCN1 in host defense against bacterial infections in animal models. Since professional phagocytes are derived from the myeloid lineage, we generated mice with *Ccn1* deleted specifically in myeloid cells by crossing *Ccn1^flox/flox* mice with the deleter strain *LysM-Cre*, yielding *Ccn1^flox/flox;LysM-Cre* mice (hereafter called *Ccn1^ΔMyeloid* mice; Supplementary Fig. 9). *Ccn1* expression in *Ccn1^ΔMyeloid* mice was greatly reduced in peritoneal macrophages, BMDMs, and BM-PMNs, but remained at WT levels in splenic B and T lymphocytes (Fig. 4a), indicating *Ccn1* deletion specifically in myeloid cells.

*S. aureus* is a leading cause of lethal bacteremia[46,47]. To examine the functions of CCN1 in bacteremia, *Ccn1^flox/flox* (control) and *Ccn1^ΔMyeloid* mice were infected intravenously (i.v.) with *S. aureus* ($7.5 \times 10^8$ CFU per mouse; Fig. 4b). Remarkably, *Ccn1^ΔMyeloid* mice exhibited 100% mortality within 5 days, whereas 60% of *Ccn1^flox/flox* mice survived at least 10 days (Fig. 4b), showing that myeloid expression of *Ccn1* strongly protects against *S. aureus* bacteremia. Bloodstream infection of *S. aureus* often disseminates to multiple internal organs, including the liver, kidney, and heart[48]. To examine tissue colonization of disseminated bacteria, we used a reduced dose of *S. aureus* ($2 \times 10^8$ CFU per mouse; i.v.) to minimize mortality (Fig. 4c, d). The liver and kidney of *Ccn1^ΔMyeloid* mice displayed higher bacterial burdens (~5-fold) compared to *Ccn1^flox/flox* mice 5 days post infection (Fig. 4c), as confirmed by H&E staining and immunohistochemistry (Fig. 4d). Both liver and kidney sections from *Ccn1^ΔMyeloid* mice displayed multiple nidi that are surrounded by pseudo-capsules and massive infiltration of neutrophils, characteristics of *S. aureus* pyogenic abscesses (Fig. 4d), which were not observed in *Ccn1^flox/flox* mice. Bacterial colonization was also much more severe in the *Ccn1^ΔMyeloid* heart, with ~100-fold more bacterial burden compared to controls (Fig. 4c).

As *S. aureus* is the most prominent cause of SSTI[49], we also examined blood dissemination in a skin infection model. After a bolus of *S. aureus* ($4 \times 10^6$ CFU per mouse) was injected subcutaneously into mouse flank skin, blood dissemination was ~2 orders of magnitude higher in *Ccn1^ΔMyeloid* mice than controls 1 day post infection, and remained high 2 days post infection while bacterial counts in *Ccn1^flox/flox* mice approached zero (Fig. 4e). Similarly, *Ccn1^ΔMyeloid* mice infected with *P. aeruginosa* showed increased mortality in bacteremia (i.v. infection) and sustained blood dissemination in peritonitis (i.p. infection), indicating that CCN1 also defends against infection by Gram-negative bacteria (Supplementary Fig. 10). These results show that *Ccn1* expression in myeloid cells, including macrophages and neutrophils, plays a crucial role in bacterial clearance in mouse models of infection.

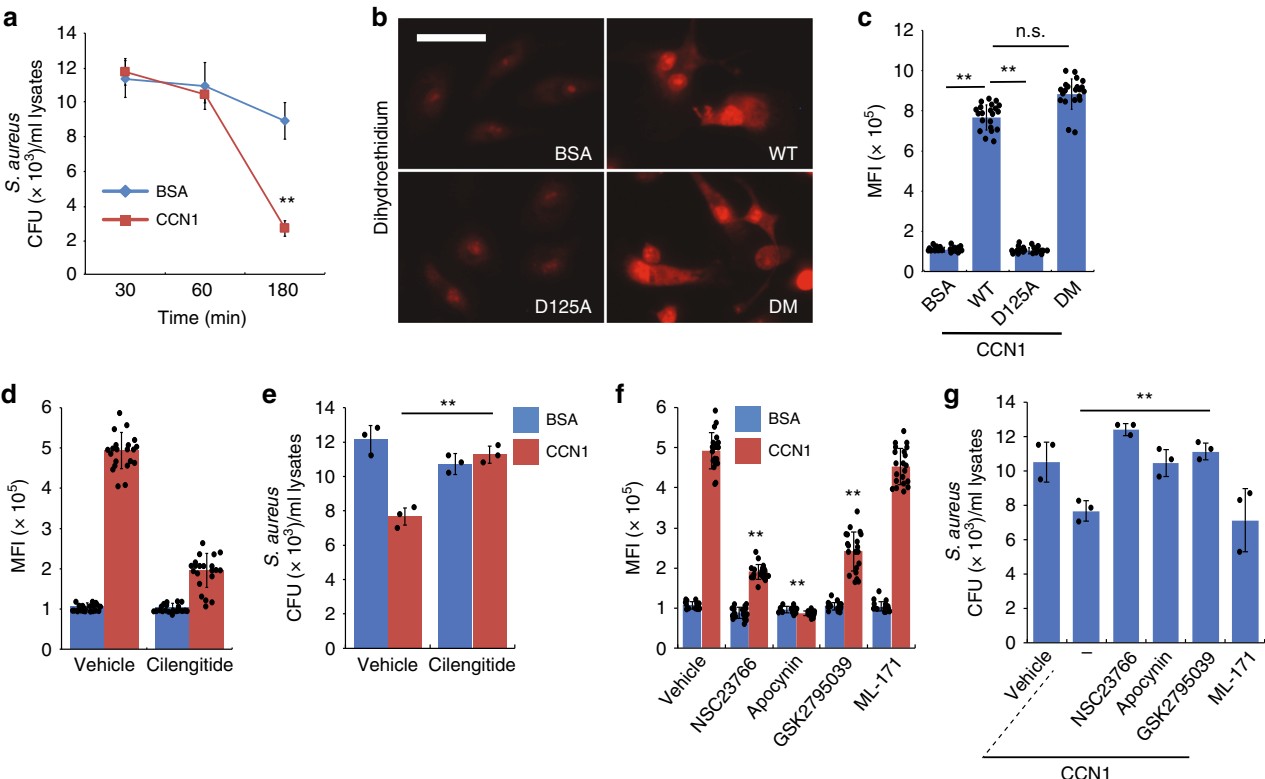

**Fig. 3 CCN1 enhances bacterial killing after phagocytosis through ROS production. a** Lysostaphin protection assay for assessing *S. aureus* killing after phagocytosis. After BMDMs phagocytosed *S. aureus* and extracellular bacteria were eliminated by lysostaphin, cells were treated with CCN1 or BSA (2 μg per ml each) and viable bacteria inside BMDMs were enumerated at indicated chase periods. **b** Superoxide ($O_2^-$) was measured by dihydroethidium (DHE, 5 μM) staining in BMDMs treated with recombinant CCN1 proteins (WT, D125A, or DM; 2 μg per ml each) for 30 min. Representative images shown were acquired from fluorescence microscopy. Bar = 40 μm. **c** High-magnification images were taken from at least ten random fields and mean fluorescence intensity (MFI) was calculated using Image J software. **d** ROS production was quantified as MFI using DHE fluorescence in BMDMs pretreated with cilengitide (1 μM), followed by CCN1 or BSA (2 μg per ml each). **e** *S. aureus* killing assays were performed as in **a** with cilengitide pretreatment. **f** The effects of various inhibitors on CCN1-induced ROS production. All inhibitors were added 30 min prior to CCN1 treatment. Rac1 inhibitor NSC23766 (10 μM); NOX inhibitor Apocynin (10 μM); NOX2 inhibitor GSK2795039 (10 μM); NOX1 inhibitor ML-171 (10 μM). **g** Effects of various inhibitors on bacterial killing using lysostaphin protection assays as above. All data were acquired from at least three independent assays and are expressed as mean ± s.d. in triplicate determinations. Statistical evaluation was performed by one-sided, two-sample with equal variance *t*-tests. **$p < 0.01$, n.s. = not significant. Source data are provided as a Source Data file.

**CCN1 drives host defense through its $\alpha_v$ integrin-binding site.** To assess the role of $\alpha_v$ integrins-mediated CCN1 phagocytic and bactericidal functions in bacterial clearance in vivo, we examined knock-in mice in which the *Ccn1* genomic locus is replaced by alleles encoding either CCN1-D125A (*Ccn1[D125A/D125A]* mice)[25] or CCN1-DM (*Ccn1[DM/DM]* mice)[26]. Upon i.v. infection with *S. aureus* ($7.5 \times 10^8$ CFU per mouse), 90% of *Ccn1[D125A/D125A]* mice perished by day 8 post infection, compared to ~20% mortality in either *Ccn1[WT/WT]* or *Ccn1[DM/DM]* mice (Fig. 5a). These results indicate that CCN1 functions mediated through $\alpha_v$ integrins, including induction of phagocytosis and ROS-dependent killing, are critical for bacterial defense in vivo. Assessment of tissue dissemination upon *S. aureus* infection ($2 \times 10^8$ CFU per mouse; i.v.) showed significantly higher levels of bacterial load in the kidney and heart from *Ccn1[D125A/D125A]* mice 5 days post infection (Fig. 5b). Interestingly, robust *S. aureus* colonization was accompanied by extensive tissue damage and loss of glomerular structures in the kidney cortex from *Ccn1[D125A/D125A]* mice, suggesting that integrin $\alpha_v$-mediated CCN1 functions may play additional roles in parenchymal injury repair (Fig. 5c).

**Exogenous CCN1 accelerates bacterial clearance.** Given the strong protective role of CCN1 against bacterial infections, we tested whether administration of exogenous CCN1 protein may

accelerate bacterial clearance in infected mice. Mice were infected with *S. aureus* ($5 \times 10^7$ CFU per mouse, i.p.) for 1 h, followed by CCN1 protein injection. Blood dissemination of *S. aureus* was observed at 3 h post infection in both *Ccn1[WT/WT]* and *Ccn1[flox/flox]* mice and was largely eliminated by 10 h (Fig. 5d, e). By contrast, high levels of *S. aureus* in blood were sustained even after 10 h in both *Ccn1[D125A/D125A]* and *Ccn1[ΔMyeloid]* mice (Fig. 5d, e). Injection of CCN1 (5 μg per mouse, i.p.) greatly reduced *S. aureus* blood dissemination, virtually eliminating bacterial counts as early as 3 h post infection in both genotypes (Fig. 5d, e). These results suggest that administration of exogenous CCN1 may have therapeutic value in the treatment of antibiotic-resistant bacterial infections.

**CCN1 binds and activates TLR2 and TLR4.** Macrophages respond to the presence of microbes by engulfment of the invaders and production of cytokines and chemokines, which marshal the infiltration of neutrophils and other immune cells for both innate and adaptive immune defense[12]. Therefore, we examined whether CCN1 may also contribute to host defense through regulating the inflammatory response. We induced acute peritonitis in mice with i.p. infection of *S. aureus* ($5 \times 10^7$ CFU per mouse) and measured the levels of inflammatory cytokines and chemokines in peritoneal lavage. Upon infection, both cytokines (TNFα and IL6) and

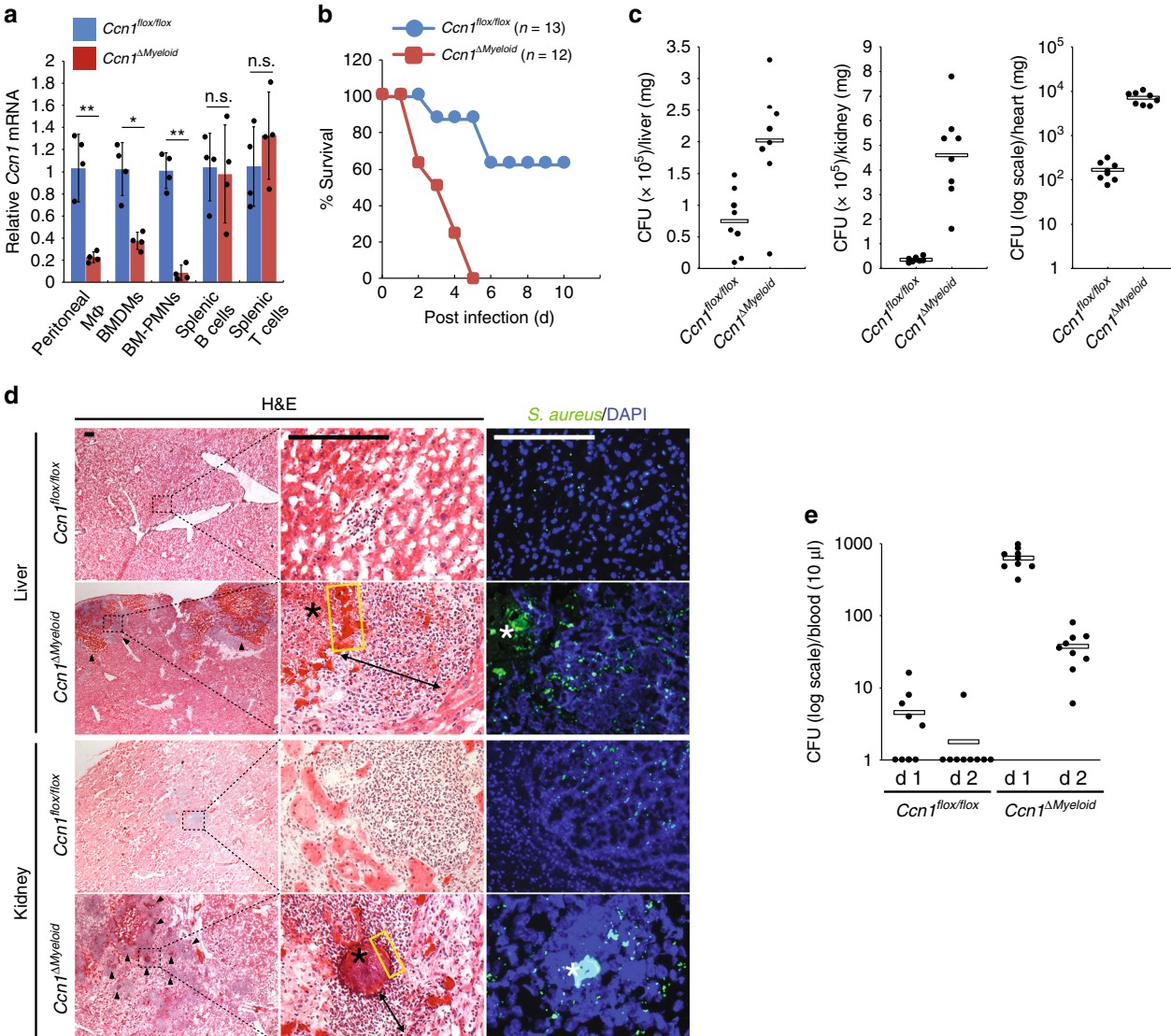

**Fig. 4 Ccn1 expression in myeloid cells is critical for host defense against S. aureus infections. a** *Ccn1* mRNA in peritoneal macrophages (MΦ), BMDMs and PMNs, and splenic B and T cells from *Ccn1^flox/flox^* and *Ccn1^ΔMyeloid^* mice was analyzed using qRT-PCR. Myeloid-specific deletion was confirmed using splenic B and T cells as controls. Data are represented as mean ± s.d. from triplicate experiments. Statistical evaluation was performed by one-sided, two-sample with equal variance *t*-tests. \**p* < 0.05, \*\**p* < 0.01, n.s. = not significant. **b** Survival of *Ccn1^flox/flox^* (*n* = 13) and *Ccn1^ΔMyeloid^* (*n* = 12) mice after infection with *S. aureus* (7.5 × 10^8 CFU per mouse, *i.v.*). **c** Tissue colonization of *S. aureus* in *Ccn1^flox/flox^* and *Ccn1^ΔMyeloid^* mice infected with *S. aureus* (2 × 10^8 CFU per mouse, *i.v.*). Bacterial counts in liver, kidney, and heart (*n* = 8 each) 5 days post infection were evaluated by growth of serially diluted tissue homogenates on tryptic soy agar with 5% sheep blood and plotted as CFU per mg tissues. **d** Histological examination of liver and kidney of *Ccn1^flox/flox^* and *Ccn1^ΔMyeloid^* mice at day 5-post infection. Representative images are shown. H&E staining showed the presence of pyogenic abscesses (arrowheads in left panels) in both liver and kidney from *Ccn1^ΔMyeloid^* mice, whereas no pyogenic abscesses were found in *Ccn1^flox/flox^* mice. Boxed regions with dotted line are expanded in high magnification in middle panels. Asterisk (\*) points to nidus, yellow rectangular box indicates pseudo-capsule, and bidirectional arrow shows regions of neutrophils. *S. aureus* colonization was visualized in adjacent sections using anti-*S. aureus* antibodies (green, Alexafluor488-anti-rabbit IgG) in immunofluorescence staining (right panels). DAPI (blue) was used as a counterstaining. Scale bar = 100 μm. **e** Blood dissemination of *S. aureus* was determined at indicated days after subcutaneous infection of *S. aureus* (5 × 10^5 CFU in 50 μl PBS) in *Ccn1^flox/flox^* and *Ccn1^ΔMyeloid^* mice (*n* = 9 each). Blood (10 μl) was plated for colony quantification. Source data are provided as a Source Data file.

chemokines (KC and MCP1) levels were significantly increased in *Ccn1^flox/flox^* mice, whereas this inflammatory response was greatly reduced in *Ccn1^ΔMyeloid^* mice (Fig. 6a–d). Consistent with decreased KC chemokine levels (Fig. 6c), complete blood count (CBC) analysis found fewer circulating neutrophils in *Ccn1^ΔMyeloid^* mice following infection (Fig. 6e). These results show that *Ccn1* expressed in myeloid cells plays a key role in the inflammatory response to bacterial infections.

To determine how CCN1 may contribute to the inflammatory response, we tested whether CCN1 can by itself induce

inflammation without infection. When *i.p.* injected into uninfected C56BL/6J (*B6*) mice, both CCN1-WT and the CCN1-D125A mutant elevated the levels of cytokines (TNFα and IL6) in peritoneal exudates and increased neutrophils in circulating blood (Supplementary Fig. 11). These results suggest that CCN1 itself, in the absence of infection, can induce sterile inflammation independent of its binding to integrin α$_v$β$_3$ and activities downstream of integrin signaling.

Sterile inflammation is often induced through TLRs recognizing DAMPs, which signal tissue damage[17,18]. Thus, we tested the

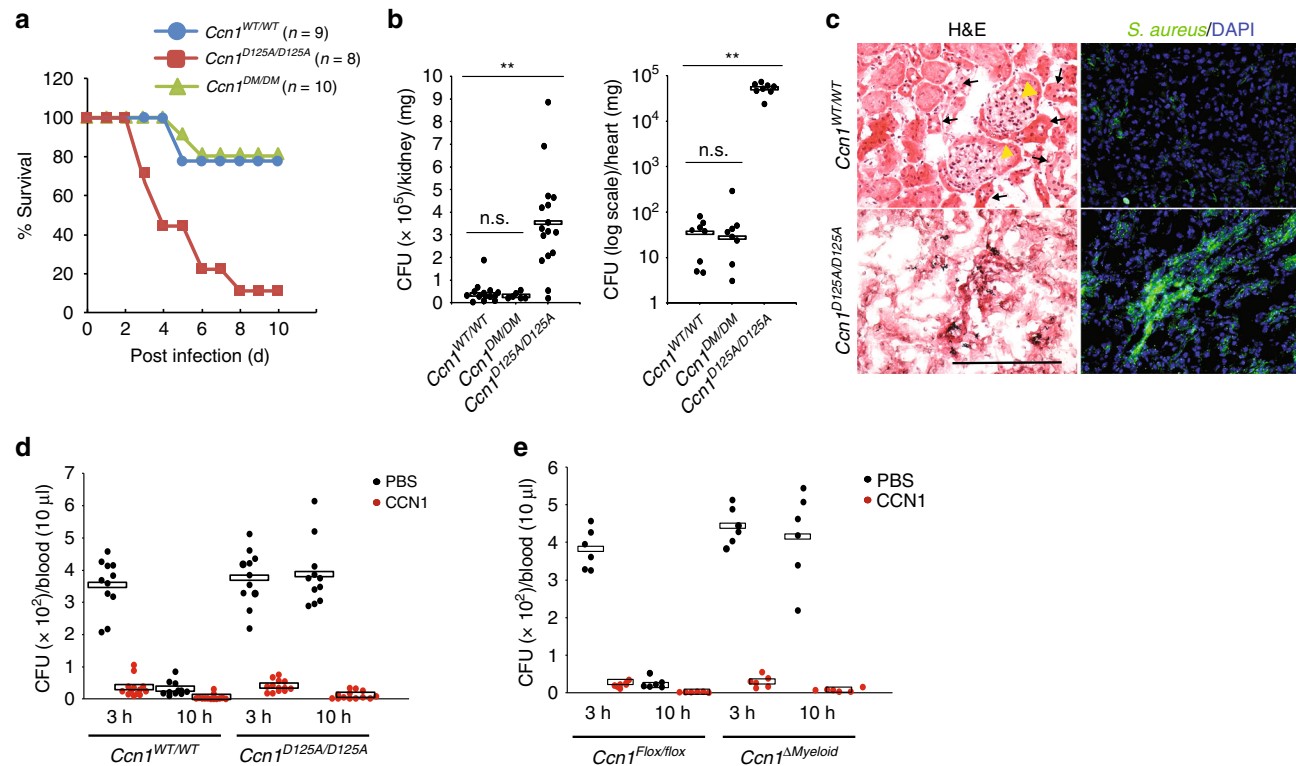

**Fig. 5 CCN1 functions mediated through its integrin $\alpha_v$-binding site are crucial for host defense. a** Survival was monitored in mice infected with *S. aureus* ($7.5 \times 10^8$ CFU per mouse, *i.v.*). $Ccn1^{WT/WT}$, $n = 9$; $Ccn1^{D125A/D125A}$, $n = 8$; $Ccn1^{DM/DM}$, $n = 10$. **b** Colonization in the kidney and heart was evaluated in mice infected with *S. aureus* ($2.5 \times 10^8$ CFU per mouse, *i.v.*). Bacterial loads were enumerated by serial dilution of tissue homogenates at day 5-post infection in $Ccn1^{WT/WT}$ ($n = 16$ for kidney, $n = 8$ for heart), $Ccn1^{D125A/D125A}$ ($n = 16$ for kidney, $n = 8$ for heart), and $Ccn1^{DM/DM}$ ($n = 6$ for kidney, $n = 8$ for heart) mice. Statistical evaluation was performed by one-sided, two-sample with equal variance *t*-tests. **$p < 0.01$; n.s. = not significant. **c** Histological evaluation of the kidney cortex after *S. aureus* infection. Kidney sections from $Ccn1^{WT/WT}$ and $Ccn1^{D125A/D125A}$ mice at day 5-post infection were stained with H&E. Glomeruli (yellow arrowheads) and tubules (black arrows) are intact in $Ccn1^{WT/WT}$ mice, but severely damaged in $Ccn1^{D125A/D125A}$ mice. Adjacent sections were visualized in immunofluorescence staining with polyclonal anti-*S. aureus* antibody (green) and counterstained with DAPI (blue). Scale bar $= 100\,\mu m$. **d** Treatment of mice with CCN1 protein accelerated bacterial clearance. $Ccn1^{WT/WT}$ and $Ccn1^{D125A/D125A}$ mice ($n = 11$ per group per genotype) were challenged with acute peritonitis by infection of *S. aureus* ($2 \times 10^7$ CFU per mouse, *i.p.*), followed by *i.p.* injection of CCN1 protein (5 µg per mouse) or PBS 1 h-post infection. Total blood was drawn at indicated times and 10 µl was plated for colony enumeration. **e** $Ccn1^{flox/flox}$ and $Ccn1^{\Delta Myeloid}$ mice ($n = 6$ per group per genotype) were infected with *S. aureus* and bacterial clearance with or without CCN1 treatment was evaluated as above. Source data are provided as a Source Data file.

possibility that CCN1 may directly regulate inflammation through TLRs using mice that lack MyD88 ($Myd88^{-/-}$), a cytoplasmic adaptor molecule that is essential for the signaling of most TLRs[50]. Remarkably, whereas *i.p.* injected CCN1 elevated the levels of both cytokines (TNFα and IL6) and chemokines (KC and MCP1) in the peritoneal lavage of *B6* mice, it failed to do so in $Myd88^{-/-}$ mice (Fig. 7a–d). Consistently, CBC analysis showed no increase in blood neutrophils in $Myd88^{-/-}$ mice upon CCN1 injection (Fig. 7e). These results indicate that CCN1-induced inflammatory response in vivo is mediated through a MyD88-dependent mechanism, possibly via TLR signaling. To assess the function of CCN1 in an isolated cell system, we treated BMDMs from *B6* and $Myd88^{-/-}$ mice with CCN1. CCN1 elevated both *Tnfa* and *Il6* mRNAs in *B6* BMDMs to levels similar to those induced by PGN, but this induction did not occur in BMDMs from $Myd88^{-/-}$ mice (Fig. 7f). As expected, induction of *Tnfa* and *Il6* expression by PGN and LPS was completely obliterated in $Myd88^{-/-}$ BMDMs, since these PAMPs are known to induce inflammatory genes through TLRs (Fig. 7f). Likewise, CCN1 also induced inflammatory gene expression in differentiated THP.1 human macrophages (Supplementary Fig. 12).

The possibility that CCN1 can activate TLR signaling suggests that it may physically interact with TLRs, among which TLR2 and

TLR4 are principal receptors for bacterial PAMPs and DAMPs[18,51]. Remarkably, we found that both human recombinant TLR2 and TLR4 proteins, which do not contain MD2, efficiently bound immobilized CCN1 in a dose-dependent manner in a solid-phase-binding assay (Fig. 8a). CD14, an LPS transport protein, failed to show binding as a negative control, indicating specificity of the assay.

To further confirm the binding of CCN1 to TLR2 and TLR4 and to assess binding affinity, we performed SPR analysis (Fig. 8b, c and Supplementary Fig. 13). Sensorgrams showed CCN1 interacted specifically with TLR2 and TLR4 immobilized on CM5 sensor chip with $K_D$ values of 227 nM and 291 nM, respectively. In addition, SPR analysis also confirmed specific interaction between CCN1 and its phagocytic receptor, integrin $\alpha_v\beta_3$, with a $K_D$ of 202 nM (Supplementary Fig. 13). These results clearly show that CCN1 binds TLR2 and TLR4 directly with affinities similar to its binding to the phagocytic receptor integrin $\alpha_v\beta_3$.

Further analysis by immunoblotting showed that CCN1-WT and CCN1-D125A bound to both TLR2 and TLR4, whereas CCN1-DM only bound TLR4 but not TLR2 (Fig. 8d). These findings indicate that the DM mutations disrupted the CCN1-binding site for TLR2. Consistent with this binding pattern,

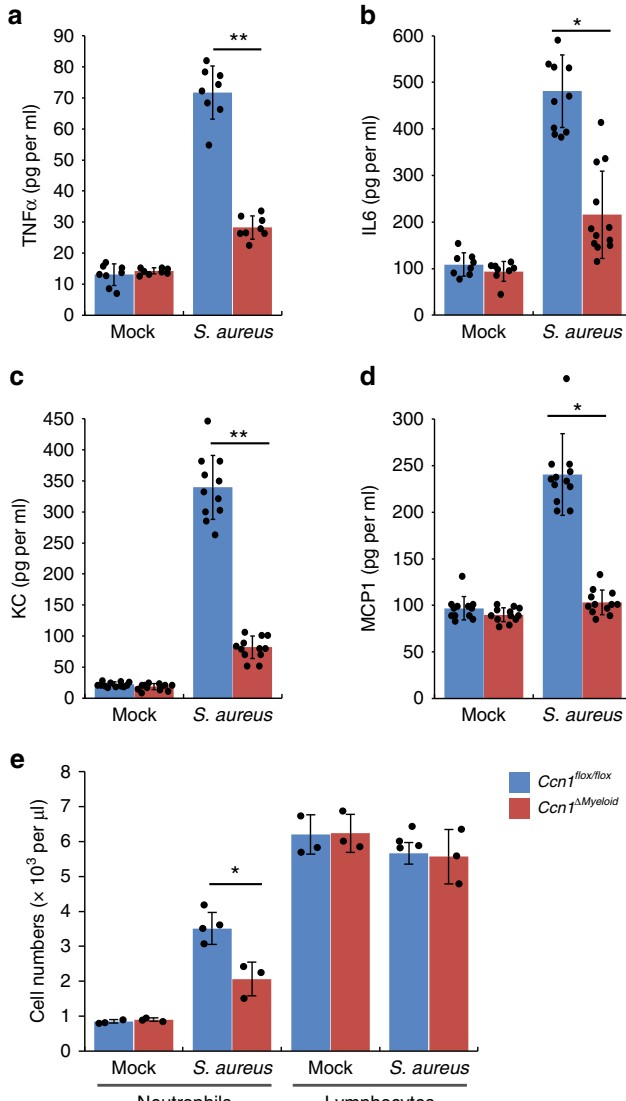

**Fig. 6 CCN1 is critical for the host inflammatory response upon bacterial infection. a–d** Quantification of cytokines and chemokines in the peritoneum in *Ccn1^flox/flox* or *Ccn1^ΔMyeloid* mice (*n* = 6 per group per genotype) infected with *S. aureus* (2 × 10^7 CFU per mouse *i.p.*). Peritoneal exudates were analyzed by ELISA to quantify the levels of TNFα (**a**) at 30 min and IL6 (**b**), KC (**c**), and MCP1 (**d**) at 2 h post infection. **e** Neutrophils and lymphocytes contents in blood drawn at 2 h post infection were determined using Advia 120 analyzer. All data are represented as mean ± s.d. acquired in triplicate determinations. Statistical evaluation was performed by one-sided, two-sample with equal variance *t*-tests. *$p < 0.05$, **$p < 0.01$. Source data are provided as a Source Data file.

CCN1-WT and CCN1-D125A induced a higher level of inflammatory gene expression in *B6* BMDMs compared to CCN1-DM, which only bound TLR4 but not TLR2 (Fig. 8e). Induction of *Tnfa* and *Il6* was abolished in BMDMs from *Myd88^−/−* mice, consistent with CCN1 activation of TLR signaling (Fig. 8e). To further confirm that CCN1 activates gene expression through direct binding to TLRs, we examined the effects of CCN1 in BMDMs from knockout mice deficient in TLR2 or TLR4. Induction of *Tnfa* and *Il6* by CCN1-WT and CCN1-D125A was reduced in BMDMs of either *Tlr2^−/−* or *Tlr4^−/−* mice, indicating that CCN1 activates gene expression through both TLR2 and TLR4 (Fig. 8f). However, CCN1-DM-induced gene expression was eliminated only in *Tlr4^−/−* BMDMs

but unchanged in *Tlr2^−/−* BMDMs (Fig. 8f), consistent with the selective binding of CCN1-DM to TLR4 but not TLR2 (Fig. 8d). Altogether, these results show that CCN1 can directly bind and activate TLR2 and TLR4 and may function as a DAMP to regulate inflammatory responses, independent of bacterial infections.

## Discussion

In a hostile environment replete with numerous and disparate microbial invaders, animal hosts have developed multiple arsenals for self-defense. Opsonins are structurally diverse frontline defense molecules that recognize and bind bacteria to mark them for phagocytosis and elimination through specific cell surface receptors[52]. Families of opsonins include antibodies, complement proteins, and factors such as pentraxins (serum amyloid P and C-reactive protein)[53,54], collectins[55], and ficolins[56]. As each opsonin may have idiosyncratic target specificities and efficacy, the nature and availability of specific opsonins at the sites of infection may affect the pathological outcome. Here, we show that the matricellular protein CCN1 is an opsonin for both Gram-positive and Gram-negative bacteria, including *S. aureus* (MRSA) and *P. aeruginosa* and promotes their removal by phagocytosis and ROS-mediated killing (Fig. 9). *Ccn1* is essential for efficient bacterial clearance in mouse models of infection, and administration of CCN1 protein markedly accelerates bacterial clearance. As the emergence of antibiotic-resistant bacteria is a critical public health issue[2,3], the efficacy of CCN1 against these common antibiotic-resistant pathogens suggests potential therapeutic value.

Our current study was predicated on the discovery that CCN1 acts as a bridging molecule for efferocytosis of apoptotic neutrophils[25]. However, most bridging molecules for efferocytosis are incapable of opsonizing bacteria, an activity that requires specific recognition of bacterial PAMPs[57]. The binding of CCN1 to PGN of *S. aureus* is mediated through both the vWC and TSR domains (Fig. 1c, g and Supplementary Fig. 3). Although thrombospondin-1 (TSP-1) has been shown to bind PGN[58], whether this binding occurs through the TSR domain of TSP-1 is unknown. The TSR of mindin binds LPS and LTA[59], whereas TSR of BAI1 binds only LPS[60]. In CCN1 the TSR binds PGN but not LPS (Supplementary Fig. 3), underscoring the functional diversity of various TSR domains. By contrast, CCN1 recognition of LPS is dependent on a cluster of positively charged residues in the CT domain (Fig. 1g)[35]. CCN1 can also bind other bacteria, including the Gram-positive *S. pneumoniae* and the intracellular Gram-negative *S. typhimurium* (Supplementary Fig. 4), indicating that CCN1 may potentially opsonize a broad spectrum of bacterial species.

Phagocytosis is a well-coordinated process involving target recognition and tethering, followed by engulfment of the cargo to form an intracellular membrane-enclosed phagosome[61]. CCN1 engages integrin $\alpha_v\beta_3$ as the phagocytic receptor, leading to the activation of the small GTPase Rac1 (Fig. 2), which is known to regulate actin cytoskeleton reorganization, cell motility, and phagocytic cup formation[62,63]. We have previously shown that in efferocytosis CCN1 activates Rac1 in macrophages upon engagement of integrin $\alpha_v\beta_3$, leading to formation of the integrin-p130^Cas/CrkII complex[25]. This complex recruits DOCK180, a Rac1-activating guanine nucleotide exchange factor (GEF), thereby activating Rac1[64,65]. Thus, CCN1 engagement of integrin $\alpha_v\beta_3$ and activation of Rac1 play crucial roles in the phagocytosis of both apoptotic cells and bacterial pathogens. Indeed, CCN1-induced Rac1 activation not only regulates phagocytic engulfment of bacteria but is also critical for NOX2-dependent ROS generation to eliminate ingested bacteria (Figs. 2 and 3), thereby parsimoniously coupling these two molecularly distinct but functionally connected events.

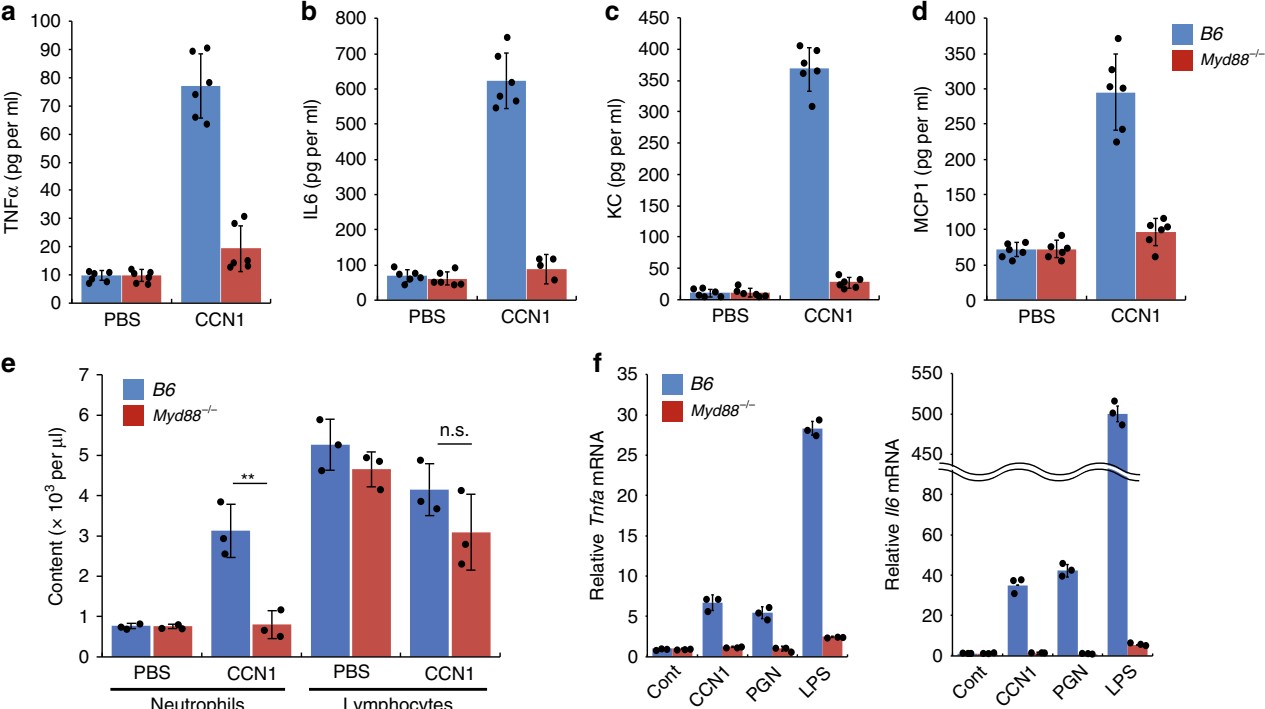

**Fig. 7 CCN1 induces *Myd88*-dependent inflammatory response. a–d** C57BL/6J (*B6*) and *Myd88*$^{-/-}$ mice (*n* = 6 each genotype) were *i.p.* injected with CCN1 protein (5 μg in 300 μl PBS), and peritoneal exudates were analyzed for TNFα (30 min) and IL6, KC, and MCP1 (2 h) using ELISA. **e** Complete blood count (CBC) analysis was performed, and neutrophils and lymphocytes contents are shown. **f** Gene expression induced by CCN1. BMDMs from *B6* or *Myd88*$^{-/-}$ mice were treated with CCN1 protein (2 μg per ml), LPS (50 ng per ml), or PGN (5 μg per ml) for 6 h. *Tnfa* and *Il6* mRNAs were quantified by qRT-PCR analyses. All data are represented as mean ± s.d. acquired in triplicate determinations. Statistical evaluation was performed by one-sided, two-sample with equal variance *t*-tests. \*\**p* < 0.01, and n.s. = not significant. Source data are provided as a Source Data file.

We have employed two mouse models, *Ccn1*$^{ΔMyeloid}$ and *Ccn1*$^{D125A/D125A}$ mice, to provide genetic evidence for the role of CCN1 in antibacterial defense in vivo (Figs. 4 and 5). First, increased mortality and tissue colonization in *Ccn1*$^{ΔMyeloid}$ mice upon *S. aureus* (MRSA USA300) and *P. aeruginosa* infection clearly demonstrate the importance of CCN1 expressed in myeloid phagocytes for efficient defense against these pathogens. Second, *Ccn1*$^{D125A/D125A}$ knock-in mice exhibited similar defects in bacterial clearance, underscoring the critical role of CCN1 activities mediated through α$_v$ integrins, including phagocytosis and generation of bactericidal ROS (Figs. 2 and 3). Interestingly, some of the distinct phenotypes of these two mouse models also hinted at cell type- and integrin-specific CCN1 functions (Figs. 4d and 5c). For example, the formation of pyogenic abscesses was prominent only in *Ccn1*$^{ΔMyeloid}$ mice but not in *Ccn1*$^{D125A/D125A}$ mice, suggesting that myeloid cells-derived CCN1 may have negative effects on abscess formation through α$_v$-independent pathways, possibly by regulating bacterial proteins important for abscess formation such as coagulase (Coa) and vWbp[66,67]. By contrast, the *Ccn1*$^{D125A/D125A}$ knock-in mice showed shrunken glomeruli and convoluted tubules of the renal cortex area upon *S. aureus* infection (Fig. 5c), suggesting that integrin α$_v$-mediated CCN1 functions in parenchymal tissues may be critical for injury repair[68]. Consistent with a role for CCN1 in parenchymal repair, CCN1 has been shown to be essential for the regeneration of bile ducts[27] and the intestinal epithelium[28] after injuries. In cutaneous wound healing, CCN1 promotes the removal of apoptotic neutrophils, accelerates wound healing progression[25], and promotes matrix remodeling[26]. The ability of CCN1 to enhance bacterial clearance may contribute to the healing of wounds at risk of infections. In this regard, *S. aureus* and *P. aeruginosa* are the two

most commonly found bacterial pathogens in chronic wounds such as diabetic wounds[69,70].

To our surprise, we found that CCN1 can by itself induce inflammatory responses in the absence of bacterial infection through physical interaction with TLR2 and TLR4 to activate MyD88-dependent signaling (Figs. 7 and 8). These findings suggest that CCN1 may act as an alarmin or DAMP to trigger sterile inflammation upon tissue injury to initiate repair. SPR analysis showed that CCN1 binds TLR2 and TLR4 with $K_D$ values of 227 nM and 291 nM, respectively. This binding is somewhat stronger than the interaction of HMGB1, a well-characterized DAMP, with the TLR4/MD2 complex ($K_D$ = 420 nM) or TLR4 alone ($K_D$ = 650 nM)[71]. Since CCN1 is released from platelet α-granules upon platelet activation[72], which is triggered by vascular damage, it can be rapidly available upon injury to induce sterile inflammation and tissue repair through direct binding to TLRs. Deregulation of CCN1 may contribute to inflammatory pathologies, as CCN1 has been associated with chronic inflammatory diseases, including rheumatoid arthritis[73], atherosclerosis[74], diabetic nephropathy and retinopathy[75,76], and inflammation-related cancers[29], and targeting CCN1 with antibodies or siRNAs ameliorates disease symptoms in animal models of rheumatoid arthritis and diabetic retinopathy[73,76].

As several previous reports on TLR-activating proteins have been confounded by the potential presence of PAMPs in the reagents used[51], we have reinforced our findings through several lines of evidence. First, our CCN1 proteins were expressed in mycoplasma-free insect cells rather than from bacterial sources. The presence of LPS was not >0.08 EU (8 pg) per μg CCN1 as judged by the Limulus Amebocyte Lysate (LAL) assay (Supplementary Fig. 14), below the amount necessary to elicit a minimal

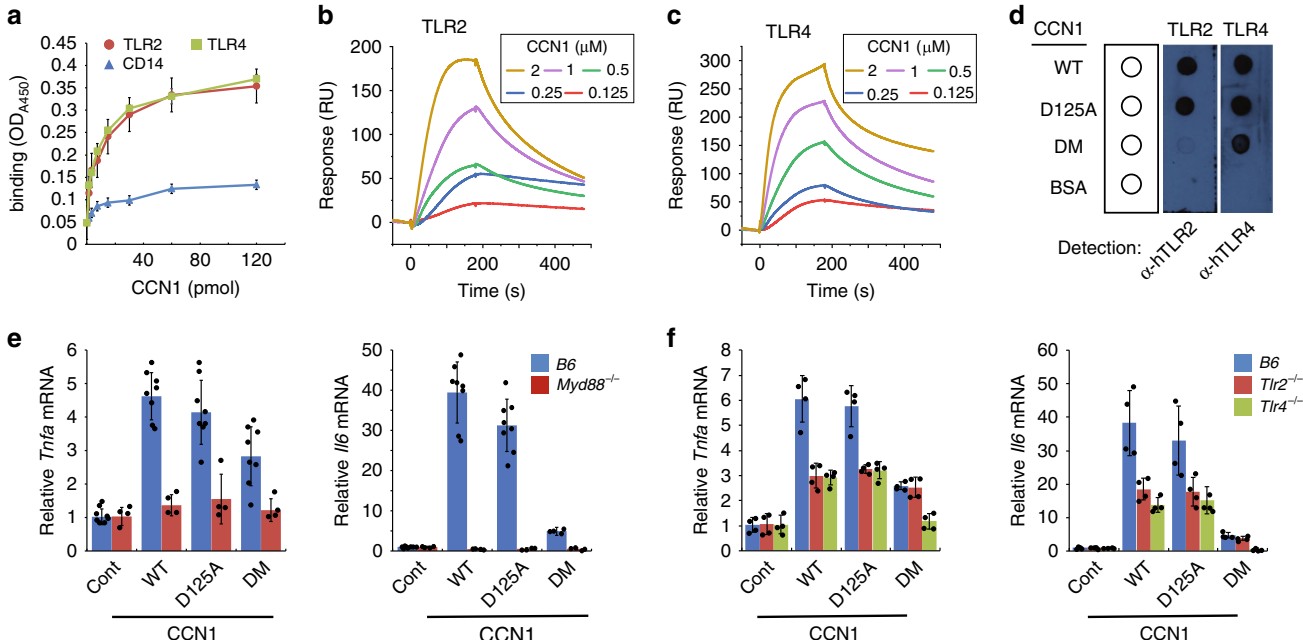

**Fig. 8 CCN1 binds and activates TLR2 and TLR4. a** Solid-phase-binding assays between CCN1 and TLR2/4. Recombinant TLR2 or TLR4 proteins (200 ng per well) were added to 96-well plates pre-coated with serially diluted CCN1 protein at indicated amounts. Specific interaction was detected and quantified using polyclonal anti-hTLR2 or anti-hTLR4 antibodies. Recombinant CD14 (200 ng per well) was used as a control. **b** SPR analyses of CCN1 binding to TLR2. TLR2 was immobilized on CM5 chip and various concentrations of CCN1 was injected as analyte. **c** Sensorgrams of CCN1 binding to TLR4 analyzed by SPR as in **b**. **d** Dot blot analyses of CCN1 and mutant proteins binding to TLR2/4. CCN1-WT, CCN1-D125A, or CCN1-DM proteins (1 μg each) were spotted onto nitrocellulose membrane and incubated with TLR2 or TLR4 proteins (2 μg each in PBS) for 4 h. Binding was detected using polyclonal anti-hTLR2 or anti-hTLR4 antibodies. A representative image is shown. **e** *Tnfa* and *Il6* mRNAs were quantified in BMDMs from either *B6* or *MyD88*$^{-/-}$ mice treated with CCN1-WT, CCN1-D125A, and CCN1-DM mutant proteins using qRT-PCR analysis. **f** *Tnfa* and *Il6* mRNAs from BMDMs of *B6*, *Tlr2*$^{-/-}$, or *Tlr4*$^{-/-}$ mice treated with CCN1 proteins (CCN1-WT, -D125A, -DM, 2 μg per ml each) were quantified using qRT-PCR analyses. All data are represented as mean ± s.d. acquired in triplicate determinations. Statistical evaluation was performed by one-sided, two-sample with equal variance *t*-tests. *$p < 0.05$, **$p < 0.01$, and n.s. = not significant. Source data are provided as a Source Data file.

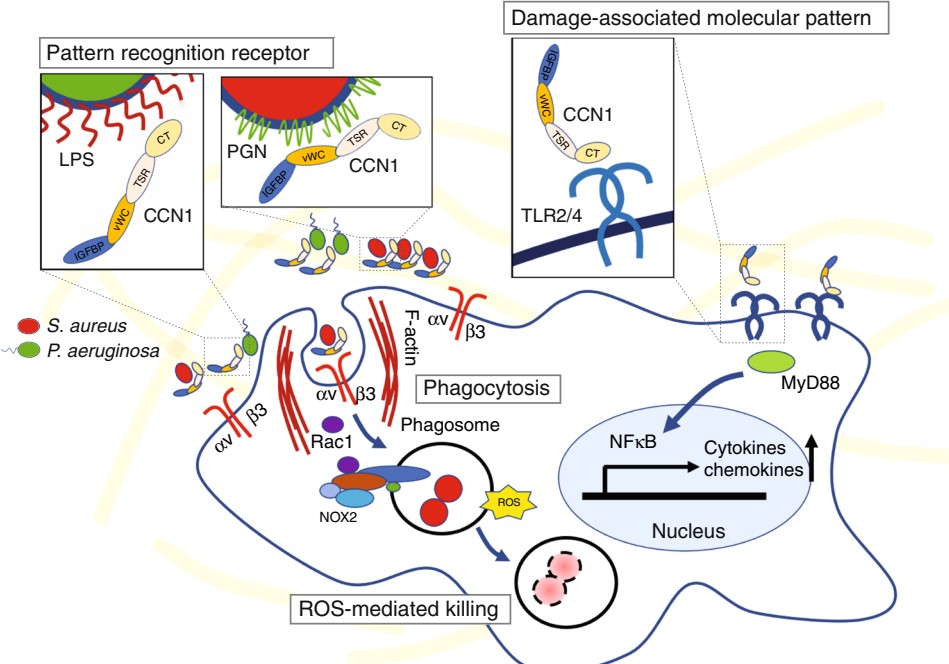

**Fig. 9 Schematic of CCN1 functions in bacterial clearance and activation of TLR signaling.** CCN1 functions as a PRR and opsonizes Gram-positive and Gram-negative bacteria through binding PGN and LPS, respectively. CCN1 activates phagocytosis by engagement of integrin $\alpha_v\beta_3$ in phagocytes, thereby promoting the engulfment of bacteria. In macrophages, CCN1 also stimulates ROS production through activation of Rac1 and NOX2, thus enhancing bacterial killing. Independent of the presence of bacteria, CCN1 functions as a DAMP and activates TLR2 and TLR4 by direct binding to these receptors, leading to MyD88-dependent expression of inflammatory cytokines and chemokines.

discernible induction of cytokine expression in various cell types[77]. Further purification through a polymyxin-B column to remove any residual endotoxin did not change its inflammation-inducing activity (Supplementary Fig. 14). Second, the CCN1-DM mutant is unable to bind LPS, yet it can bind TLR4 and induce TLR4 signaling (Figs. 1g and 8f), indicating CCN1 activation of TLR is independent of LPS binding. Further, CCN1-DM is defective for binding TLR2 and is correspondingly unable to induce TLR2 signaling, indicating that TLR binding and activation require specific sequences in the CCN1 polypeptide (Fig. 8d, f). Finally, genetic evidence from $Ccn1^{\Delta Myeloid}$ and $Myd88^{-/-}$ mice showed that CCN1 contributes to inflammatory responses through MyD88, supporting its functions through TLRs in vivo (Figs. 6 and 7). Altogether, these observations provide compelling evidence that CCN1 acts as a TLR-activating DAMP.

*S. aureus* and *P. aeruginosa* have developed diverse mechanisms of immune evasion by interfering with host opsonins, particularly those that are constitutively expressed[78,79]. For example, mucoid exopolysaccharides from *P. aeruginosa* interferes with complement-dependent phagocytosis[80], and protein A of *S. aureus* binds to the Fc region of IgG, thereby blocking FcR-dependent phagocytosis[81,82]. However, it appears that neither *S. aureus* nor *P. aeruginosa* has the ability to counteract the opsonin activities of CCN1, as demonstrated by results in *Ccn1* genetic models and the clear efficacy of exogenous CCN1 in promoting clearance of these bacteria (Figs. 4 and 5, and Supplementary Fig. 10). Furthermore, the activity of CCN1 as an opsonin should be indifferent to the antibiotic resistance status of the bacterial strains. Thus, CCN1 may potentially be useful therapeutically against a broad range of Gram-positive and Gram-negative antibiotics-resistant bacteria. Further studies that define the minimal functional domains of CCN1 for opsonization and phagocytosis may allow the construction of peptides that link these domains for bacterial clearance. Future investigation into the potential efficacy of CCN1 and CCN1-derived peptides may prompt novel therapies for bacterial infections recalcitrant to conventional treatments.

## Methods

**CCN1 proteins and reagents**. Recombinant CCN1 and mutant proteins (CCN1-D125A, CCN1-DM, CCN1-ΔCT, and IGFBP-vWC; Supplementary Fig. 1) were produced using a baculovirus expression system in Sf9 insect cells and purified by ion-exchange or immuno-affinity chromatography[34,36]. Sf9 cells were routinely tested for mycoplasma contamination using LookOut® Mycoplasma PCR detection kit (Sigma, MP0035). CCN1 preparations were tested for the presence of LPS using Limulus Amebocyte Lysate (LAL) Chromogenic Endotoxin Quantification kit (Thermo Scientific, 88282) and found not >0.08 EU per μg CCN1 (Supplementary Fig. 14A). Further purifications through a polymyxin-B-agarose column (Sigma, P1411; 1 mg per ml; binding capacity of 200–500 μg LPS per 1 mg) to remove any endotoxin did not eliminate CCN1 (Supplementary Fig. 14B). Individual CCN1 domain fragments (IGFBP, vWC, TSR) were produced as GST-fusion proteins and purified using a glutathione-Sepharose column[83]. Additionally, CCN1 proteins were purchased from R&D biosystems (4055-CR-050) and Novus (NBP2-34944) to test CCN1 activities (Supplementary Fig. 14). Rabbit and goat IgGs were purchased from Fisher Scientific. LPS (L9143) from *P. aeruginosa*, PGN (77140) and LTA (L2515) from *S. aureus*, Rac1 inhibitor (NSC23766; 553502), PI3K inhibitor (LY294002; 19-142), ERK inhibitor (PD98059; 19-143), p38 MAPK inhibitor (SB203580; S8307), cytochalasin D (C2618), and Apocynin (178385) were from Sigma. NOX1 inhibitor (ML-171; 492002) was from EMD Millipore. NOX2 inhibitor (GSK2795039; HY-18950) was from MedChemExpress. Cilengitide (S7707) was from SellecChem.

**Bacterial culture**. All bacterial stains used were obtained from ATCC. Methicillin-resistant *S. aureus* strain USA300 (ATCC®-BA1717™), *P. aeruginosa* (ATCC®-27107™), *S. pneumoniae* (ATCC®-6303™), and *S. typhimurium* (ATCC®-14028™). *S. aureus* and *P. aeruginosa* were cultured overnight in tryptic soy broths, diluted at 1:100 and grown exponentially. In general, $OD_{A600} = 1$ corresponded to $1.5 \times 10^9$ CFU per ml for *S. aureus* and $7.5 \times 10^8$ CFU per ml for *P. aeruginosa*. Bacteria were used at their exponential growth phase ($OD_{A600} = {\sim}0.6$). *S. pneumoniae* and *S. typhimurium* were grown in Brain-Heart infusion broth (Sigma, 53286).

**Animals and bacterial infection models**. Animal protocols were approved by the Institutional Animal Care and Use Committee of The University of Illinois at Chicago (ACC#17-001). $Ccn1^{D125A/D125A}$ and $Ccn1^{DM/DM}$ knock-in mice[25,26] were generated in a svJ129-C57BL/6J mixed background and backcrossed to the C57BL/6J strain 6 and 11 times, respectively. $Ccn1^{\Delta Myeloid}$ mice were generated by crossing $Ccn1^{flox/flox}$ with myeloid-specific *Cre* deleter stain, *LysM-Cre* (B6.129P2-*Lyz2^{tm1(cre)Ifo}*/J, 004781). $Tlr2^{-/-}$ (B6.129-*Tlr2^{tm1Kir}*/J, 004650), $Tlr4^{-/-}$ (B6(Cg)-*Tlr4^{tm1.2Karp}*/J, 029015), and $Myd88^{-/-}$ mice (B6.129P2(SJL)-*Myd88^{tm1.1Defr}*/J),009088) were from the Jackson Laboratory. All mice were housed in sterile static microisolator cages on autoclaved corncob bedding with water bottles. Both irradiated food and autoclaved water were provided ad libitum. The standard photoperiod was 14 h of light and 10 h of darkness. Mice (both male and female) at 10–12 weeks of age with similar body weight (25–28 g) were used in bacterial infection studies. For bacteremia and tissue colonization studies, mice were anesthetized by intraperitoneal (i.p.) injection of ketamine (100 mg per kg) and xylazine (10 mg per kg) and systemically infected with *S. aureus* ($7.5 \times 10^8$ CFU per mouse for survival, $2 \times 10^8$ CFU per mouse for tissue colonization) through retro-orbital delivery. Liver (whole-left lateral lobe), kidney (two whole-organs per mouse) and the whole heart were harvested and processed for bacterial colonization and histological analyses. For acute peritonitis model, mice were infected by i.p. injection of *S. aureus* ($5 \times 10^7$ CFU per mouse) or *P. aeruginosa* ($5 \times 10^7$ CFU per mouse) and peritoneal exudate or whole blood were examined for bacterial blood dissemination and cytokines/chemokines production. In CCN1 protein injection studies, purified CCN1 (5 μg in PBS) was i.p. injected in animals to analyze inflammatory responses or injected 1 h after *S. aureus* infection to monitor bacterial clearance. For skin and soft-tissue infection (SSTI), a bolus of *S. aureus* ($4 \times 10^6$ CFU per mouse) was subcutaneously injected into mouse flank skin and blood dissemination was monitored.

**Complete blood count**. Mice with acute peritonitis had total blood drawn via cardiac puncture and complete blood count (CBC) was analyzed using Advia 120 analyzer (Siemens).

**Cell culture and isolation**. All cell lines were obtained from ATCC. L-929 cells (ATCC®CCL-1™) were grown in Dulbecco's modified Eagle medium (DMEM) Glutamax I media (Invitrogen) supplemented with 10% (vol per vol) heat-inactivated fetal bovine serum (FBS, Hyclone™), 1% HEPES, and 1% penicillin-streptomycin (Invitrogen) in a humidified incubator with 5% $CO_2$ at 37 °C. For preparation of conditioned medium, L-929 cells were plated at density of $4.7 \times 10^5$ cells in a 75-cm² flask containing 55 ml culture medium and grown for 7 days. Collected supernatant was filtered through a 0.45 μm filter and stored at –30 °C. THP.1 cells (ATCC®TIB-202™) were grown in suspension in RPMI Glutamax media (Invitrogen) supplemented with 10% FBS and 0.05 mM β-mercaptoethanol. For differentiation, THP.1 cells at low passage (<15) were plated in 4-well chamber slide in the presence of phorbol-12-myristate-13-acetate (PMA, 100 nM; Sigma) for 3 days. Peritoneal macrophages were isolated from peritoneal exudates of mice elicited with thioglycolate (3% vol per weight) for 3 days[84]. Bone marrow-derived neutrophils (PMNs) were isolated by differential centrifugation[25]. The purity of the neutrophil fraction was assessed by Giemsa staining (Sigma); over 99% purity was typically obtained. For BMDMs, the total bone-marrow cells were re-suspended in macrophage complete media (DMEM/F12 with 10% FBS and 20% L-929 cells conditioned media), and $4 \times 10^5$ cells were plated in sterile plastic petri dish (100 mm) and incubated in 5% $CO_2$ at 37 °C for 7 days with medium being refreshed every 3 days for differentiation into macrophages. Macrophage differentiation was assessed by F4/80 immunostaining; typically, over 90% are F4/80 positive. For splenic B or T cells isolation, total spleen was excised and homogenized between frosted ends of microscope slides in Hank's Balanced Salt Solution. Subsequently, B or T cells were isolated using Nylon Wool Fiber Columns (Polysciences) according to manufacturer's instruction.

**Bacterial phagocytosis and killing**. For *S. aureus* phagocytosis, BMDMs were labeled with CellTracker™ Green-CMFDA (5-chloromethylfluorescein diacetate) fluorescent dye (5 μM, Invitrogen) for 30 min. Cells were either untreated or pretreated with CCN1 proteins (CCN1-WT, -D125A, and -DM; 2 μg per ml each) for 1 h, followed by incubation with *S. aureus* bioparticle (pHrodo® Red *S. aureus* BioParticles™ conjugates; Invitrogen, 5 μg per ml) for an additional 45 min. Cells were washed with ice-cold PBS five times and fixed with paraformaldehyde (4% vol per vol in PBS). Images were taken using fluorescence microscopy (Leica DM4000B). Phagocytosis was quantified by counting macrophages containing >2 *S. aureus* particles in 12 randomly selected high-powered fields and expressed as a percentage of positive macrophages over total cells counted (~300 cells). For all inhibitor experiments, cells were pretreated with chemical inhibitors for 30 min prior to CCN1 treatment. To monitor bacterial killing after phagocytosis, BMDMs were allowed to phagocytose *S. aureus* (MOI of 10) for 90 min, followed by CCN1 treatment with or without chemical inhibitors. To remove unphagocytosed *S. aureus*, lysostaphin (10 μg per ml; Sigma) was added for 15 min. Cells were lysed in Triton X-100 (0.04% vol per vol) and plated with serial dilution on tryptic soy agar plates with 5% sheep blood for bacterial enumeration.

To monitor phagocytosis of *P. aeruginosa*, BMDMs were pretreated with CCN1 proteins, followed by incubation with *P. aeruginosa* (MOI of 10) for 45 min. Cells were treated with gentamycin (200 µg per ml; Gibco) for 15 min to eliminate extracellular or membrane-bound bacteria, lysed in Triton X-100 (0.04% vol per vol), and plated with serial dilution on tryptic soy agar with 5% sheep blood for bacterial enumeration.

**ROS measurement**. For measurement of superoxide radicals ($O_2^-$), BMDMs were grown in 4-well chamber slides and pre-loaded with dihydroethidium (DHE, 5 µM; Invitrogen) in serum-free media for 1 h. Cells were then treated with CCN1 proteins in full growth media for an additional 45 min and washed with ice-cold PBS three times. High-resolution images were taken from randomly selected fields using fluorescence microscopy (Leica DM4000B) and fluorescence intensity of individual cells (>50 cells) was analyzed using Image J (NIH). Chemical inhibitors were treated 30 min prior to CCN1 proteins. Cells were kept in darkness during the entire procedure.

**Solid-phase-binding assay and dot blot analysis**. The physical interaction of CCN1 with bacteria (*S. aureus* or *P. aeruginosa*) or TLRs was analyzed in solid-phase-binding assays. Ninety-six-well plates were coated with serially diluted CCN1 or other ECM proteins in bicarbonate/carbonate coating buffer (100 mM, pH 9.6) overnight at 4 °C. After washing three times with PBS-Tween20 (PBST), plates were then blocked with poly-vinyl alcohol (PVA, 2% vol per weight) for 1 h. For bacterial binding, *S. aureus* or *P. aeruginosa* ($8 \times 10^3$ CFU per well in 100 µl PBS) was added to each well and incubated for 1 h. After washing with PBST three times, bound bacteria were detected using polyclonal anti-*S. aureus* (abcam, ab20920; 1:2000 dilution) or anti-*P. aeruginosa* antibodies (abcam, ab69232; 1:500 dilution), visualized with horseradish peroxidase (HRP)-conjugated secondary antibodies with 3,3′,5,5′- Tetramethylbenzidine (TMB) as a substrate, and quantified in a microplate reader (Labsystems Multiskan MS) at A450 nm. Absorbance at A570 nm was used to correct for minor optical imperfections of plates. All assays were done in triplicates and in three independent experiments. For TLRs interaction, hTLR2, hTLR4, or hCD14 (0.2 µg per well, R&D systems) were incubated with coated CCN1 proteins and detected using polyclonal anti-hTLR2 (AF2616), anti-hTLR4 (AF1478), or monoclonal anti-hCD14 antibodies (MAB3832, R&D systems; 1:100 dilution each,) as described above. In some cases, interactions were assayed in dot blot analyses. Briefly, bacterial pattern molecules (PGN, LTA, or LPS; 1 µg each) were spotted on nitrocellulose membranes and air-dried for 30 min. Each membrane was incubated with CCN1 proteins (CCN1-WT, -D125A, or -DM proteins; 5 µg in 5 ml PBS each) and bound CCN1 was detected with polyclonal anti-CCN1 antibodies (1:2000 dilution) using chemiluminescence. For TLRs binding, CCN1 proteins (CCN1-WT, -D125A, and -DM; 1 µg each) were spotted, followed by incubation with hTLR2 or hTLR4 (each 5 µg in 5 ml PBS). Bound TLRs were detected with polyclonal anti-hTLR2 or anti-hTLR4 antibodies (1:1000 dilution each).

**Flow cytometry**. *S. aureus* collected in the logarithmic phase were heat-killed in 80 °C for 30 min and re-suspended in PBS buffer containing 2 mM EDTA and 0.5% BSA. *S. aureus* ($10^8$ CFU per ml) were then incubated with increasing concentrations of either CCN1 or recombinant human C3b complement proteins (Sigma, 204860) for 1 h at room temperature. After washing once with buffer, rabbit polyclonal anti-CCN1 antibody (1:100 dilution) or mouse monoclonal anti-iC3b antibody (EMD Millipore, MABF972; 1:200 dilution) were added for 1 h, followed by allophycocyanin (APC)-conjugated goat anti-rabbit IgGs (Life Technologies Co.; A10931; 1:250 dilution;) or APC-conjugated rat anti-mouse IgGs (Life Technologies Co.; 17-4015-82, 1:500 dilution), respectively. *S. aureus* was then analyzed using CytoFLEX S flow cytometer (APC channel, Beckman Coulter) and histograms were generated using CytoExpert software. The gating strategy is shown in Supplementary Fig. 2.

**SPR analysis**. SPR studies were performed using a Biacore T-200 (Biacore Inc., GE Healthcare) biosensor system according to the manufacturer's instructions. TLR2, TLR4 and integrin $\alpha_v\beta_3$ proteins were directly immobilized on a CM5 sensor chip using the standard amine-coupling method. The carboxymethylated dextran surface was activated by 1-ethyl-3-(3-dimethylaminoprophyl) carbodiimide hydrochloride (EDC)/N-hydroxy succinimide (NHS) mixture. TLR2 (2616-TR-050, R&D systems), TLR4 (1478-TR-050) and integrin $\alpha_v\beta_3$ proteins (3050-AV-050) were diluted with 10 mM sodium acetate buffers (pH 4.0) and immobilized on respective flow channels, followed by ethanolamine blocking of the unoccupied surface area. As an analyte, various concentrations of CCN1 were applied at 25 µl per min flow rate at 25 °C in phosphate buffer containing 350 mM sodium chloride, 0.5 mM calcium chloride and 0.5 mM magnesium chloride and 10 µg per ml BSA. LPS was immobilized on a hydrophobic HPA sensor chip and CCN1 protein was used as analyte. For immobilization, the HPA surface was washed with a 5-min injection of 40 mM octyl glucoside in water, followed by LPS injection at a low flow rate (5 µl per min). Loosely bound LPS was then removed by washing the surface with a short injection (1 min) of 100 mM sodium hydroxide. The instrument was kept completely detergent-free and buffers were thoroughly degassed for experiment with HPA chip as recommended. Data (response unit, RU) were

double subtracted by blank reference channel and zero concentration analyte signal. Sensorgrams were analyzed using the Biacore T-200 evaluation software V.3.0 and kinetic rate constants ($k_a$ and $k_d$) were determined by fitting globally to the 1:1 Langmuir (one-to-one binding) model embedded within the software. The equilibrium dissociation constant ($K_D$) was calculated as the ratio of these two constants ($k_d/k_a$).

**Tissue histology**. Liver and kidney tissues from mice at 5 day post infection were snap-frozen in optimum cutting temperature (OCT) compound (Tissue-Tek) and sectioned serially (7 µm) using a Leica CM1950 UV cryostat. Sections fixed with cold acetone were stained with Hematoxylin/Eosin (H&E) or probed with polyclonal anti-*S. aureus* antibodies (abcam; ab20920, 1:100 dilution) in immunofluorescence staining with Alexafluor488-conjugated anti-rabbit IgG (Invitrogen, 1:500 dilution). DAPI was used as the counterstain (1 mg per ml, Sigma). Fluorescence images were acquired using a Leica DM4000B microscope and processed with Photoshop CC 2019 (Adobe).

**Knockdown of integrin subunit in BMDMs**. Predesigned Dicer-Substrate Short Interfering RNAs (DsiRNAs, 2 nM; Integrated DNA Technologies) against specific integrin subunits were transfected into BMDMs using SuperFect Transfection reagent (Qiagen) according to manufacturer's manuals. Knockdown efficiencies (>90%) were confirmed using quantitative reverse transcription PCR (qRT-PCR) analyses after 24 h. DsiRNA sequences used for knockdown are shown in Supplementary Table 1.

**Quantification of cytokines and chemokines**. Mice were challenged with acute peritonitis induced by either *S. aureus* infection or CCN1 injection. Peritoneal exudates were harvested with 4 ml ice-cold PBS using 18-gauge needles; cytokines and chemokines were quantified using ELISA kits: mTNFα (88-7324-22, eBioscience), mIL6 (88-7064-22, eBioscience), KC (EMCXCL1, Invitrogen), and mMCP-1 (88-7391-22, Fisher Scientific).

**RNA isolation and qRT-PCR**. Cultured BMDMs were homogenized using TRIzol reagent (Invitrogen) and total RNAs were purified using RNeasy® Mini Kit (Qiagen). Total RNAs (2 µg) were reverse transcribed to complementary DNA using MMLV-Reverse Transcriptase (Promega), and qRT-PCR was performed with the iCycler Thermal Cycler (Bio-Rad) using iQ SYBR Green Supermix (Bio-Rad). The specificity of qRT-PCR was confirmed by agarose gel electrophoresis and melting curve analysis. A housekeeping gene (*Cyclophilin E*) was used as an internal control. The primers used are following: *Tnfa*, forward 5′-CATCTTCTCAAAATTCGAGTGACAA-3′, reverse 5′-TGGGAGTAGACAAGGTACAACCC-3′; *Il6*, forward 5-GAGGATACC ACTCCCAACGAGCC-3′, reverse 5′-AAGTGCATCATCGTTGTTCATACA-3′; *Cyclophilin E* (*CypE*), forward 5′-TTCACAAACCACAATGGCACAGGG-3′, reverse 5′-TGCCGTCCAGCCAATCTGTCTTAT-3′.

**Statistics**. All experimental results are expressed either as mean ± standard deviation (s.d.) or mean with individual quantitative values in dot plots. Wherever necessary, statistical evaluation was performed by one-sided, two-sample with equal variance *t*-tests. A $p < 0.05$ value was considered significant. All quantitative experiments were performed in triplicates unless otherwise indicated.

**Reporting summary**. Further information on research design is available in the Nature Research Reporting Summary linked to this article.

## Data availability

Data supporting the findings of this work are available within the paper and the Supplementary Information files. The raw data are provided in Source Data file or from the corresponding author upon request.

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

## Acknowledgements

We thank Dr. Hyun Lee of the UIC Biophysics Core for help with SPR analysis and Seung Won Shin for excellent assistance. This work was supported by grants from the National Institutes of Health (AR061791, GM078492, and DK108994) to L.F.L.

## Author contributions

J.-I.J. conducted the experiments; J.-I.J. and L.F.L. designed the experimental plan, analyzed the data, and wrote the paper.

## Competing interests

The authors declare no competing interests.
