## [Peer Review File · Nature Communications]

Reviewers' comments:

Reviewer #1 (Remarks to the Author):

This paper is written by recognized experts in CCN1 biology. The authors extend their previous work stating that exogenously added recombinant CCN1 induces efferocytosis of apoptotic neutrophils. Here, they show rCCN1 binds *S. aureus* and *P. aeruginosa* and LPS. Engulfment of *S. aureus* by macrophages is enhanced by CCN1. , DsiRNA-mediated knockdown of either integrin α v (Itgav) or β 3 (Itgb3) efficiently blocked CCN1-induced phagocytosis, whereas knockdown of α 6 (Itga6), α M (Itgam), or β 2 (Itgb2) did not. Inhibition of Rac1 by NSC23766 blocked CCN1-induced phagocytosis. CCN1 opsonizes both Gram-positive and Gram-negative bacteria for phagocytosis, using integrin α v β 3. CCN1 accelerated bacterial killing after phagocytosis. Rac1 inhibitor NSC23766 efficiently blocked ROS production and bacterial killing upon CCN1 treatment. Results were backed up using animal models using mutated CCN1.. Exogenous CCN1 protein accelerated bacterial clearance in infected mice. CCN1 can directly bind and activate TLR2 and TLR4. These data are novel and reveal a new and interesting function for CCN1

Comments

1 "Recombinant CCN1 and mutant proteins (CCN1-D125A, CCN1-DM, CCN1- Δ CT, and IGFBPvWC) were produced using a baculovirus expression system in Sf9 insect cells and purified by ion-exchange or immuno-affinity chromatography"

Can the authors provide information as to the exact amino acids involved in these fragments, and how the authors assessed purity. CCN proteins are "sticky" how are the authors positive that the CCN preparations are contaminant-free (ie free of additional proteins). It would be difficult at present for other authors to duplicate these data given the proprietary homemade reagents

2. please provide catalog #s of inhibitors

3. siRNA studies, Was only one siRNA used? It does not appear that westerns were used to assess knockdown

4. can the authors specifically comment on the role of endogenous full length CCN1 versus exogenously added CCN1 in the activities reported here. Are the levels of exogenous CCN1 physiologically relevant? Please comment on how to advance these ideas therapeutically eg regarding CCN1 domains or peptides

Reviewer #2 (Remarks to the Author):

This paper aims to show that CCN1 is an opsonin for Gram negative as well as Gram positive bacteria.

The authors claim that CCN1 bind to LPS, peptidoglycan, TLR2, TLR4 and as well to α v β 3.

This would be a highly remarkable and therefore such statement should be made in the context of proper observations with solid binding studies. To my surprise the authors only show immunoassay results. They need to show affinity for all of these ligands, generated by SPR, ITC or similar solid binding-analysis. Also for the purpose of estimating how much CCN1 binds they should compare binding of CCN1 on bacteria (flow cytometry) to established opsonins as antibodies and complement. How much is binding and how strong is it binding.

Proper opsonins only bind to cells when bound to bacteria (as antibodies and complement do)

Otherwise this protein would always activate TLR2 and 4. Authors should show What is the mechanism of this, and do provide data to show this

Same hold true for interaction with α v β 3.

Please provide data on purity of protein, including proper endotoxin measurements. I am not convinced by what is reported.

Although most of the mice data are solid, this paper needs mechanistic experiments to make the claims that the authors try to make.

Reviewer #3 (Remarks to the Author):

Jun and Lau investigate the ability of CCN1, a matricellular protein known to be involved in a number of key host functions, to function as a host opsonin and elicit components of the inflammatory response. The authors ultimately use mouse models to demonstrate that CCN1 has an important role in the host innate immune response. Overall, the work is interesting and a significant advance for the field. I have a few comments for the authors to consider.

1. It would be optimal to measure production of extracellular superoxide by reduction of cytochrome c, a traditional quantitative assay (see Bylund et al., *Methods Mol Biol.* 2014;1124:321-38). This would provide information about the quantity of superoxide produced by macrophages (e.g., as in Figure 3). This assay is superior to the DHE assay for quantitation and measures extracellular superoxide—which I presume is the process for cells activated by CCN1.
2. Protection related to promoting phagocytosis of *S. aureus* in the mouse model is not likely to be as significant in humans. This is because laboratory mice typically lack *S. aureus*-specific antibody, whereas humans contain antibody specific for *S. aureus*. Most notably, *S. aureus* is ingested readily by human phagocytes (neutrophils) in whole blood or in vitro following opsonization with normal human serum. Thus, *S. aureus* vaccine approaches that are directed to promote opsonophagocytosis often work well in mouse models, but fail in human systems, including past clinical trials. The ability of CCN1 to activate TLR2 and TLR4 is perhaps more important in context of a potential therapeutic for *S. aureus* infections. These points should be considered by the authors—probably in the discussion section.
3. It would be optimal to test/verify the ability of CCN1 to function with human phagocytic cells in vitro, and/or elicit inflammatory responses with human cells in vitro.

We appreciate the reviewers' constructive comments and helpful suggestions. Our responses to each comment are highlighted below in purple.

Reviewer #1 (Remarks to the Author):

This paper is written by recognized experts in CCN1 biology. The authors extend their previous work stating that exogenously added recombinant CCN1 induces efferocytosis of apoptotic neutrophils. Here, they show rCCN1 binds *S. aureus* and *P. aeruginosa* and LPS. Engulfment of *S. aureus* by macrophages is enhanced by CCN1. DsiRNA-mediated knockdown of either integrin α v (Itgav) or β 3 (Itgb3) efficiently blocked CCN1-induced phagocytosis, whereas knockdown of α 6 (Itga6), α M (Itgam), or β 2 (Itgb2) did not. Inhibition of Rac1 by NSC23766 blocked CCN1-induced phagocytosis. CCN1 opsonizes both Gram-positive and Gram-negative bacteria for phagocytosis, using integrin $\alpha\beta$ 3. CCN1 accelerated bacterial killing after phagocytosis. Rac1 inhibitor NSC23766 efficiently blocked ROS production and bacterial killing upon CCN1 treatment. Results were backed up using animal models using mutated CCN1. Exogenous CCN1 protein accelerated bacterial clearance in infected mice. CCN1 can directly bind and activate TLR2 and TLR4. These data are novel and reveal a new and interesting function for CCN1

Comments

1. "Recombinant CCN1 and mutant proteins (CCN1-D125A, CCN1-DM, CCN1- Δ CT, and IGFBP-vWC) were produced using a baculovirus expression system in Sf9 insect cells and purified by ion-exchange or immuno-affinity chromatography." Can the authors provide information as to the exact amino acids involved in these fragments, and how the authors assessed purity.

As suggested, we have incorporated information on the specific amino acids involved in the integrin-binding mutants and protein fragments in Supplementary Fig. S1.

We routinely assessed the purity of recombinant proteins by SDS-gel electrophoresis, which allowed us to estimate that the protein preps are >90-95% pure. Please also see below regarding protein purity.

2. CCN proteins are "sticky" how are the authors positive that the CCN preparations are contaminant-free (ie free of additional proteins). It would be difficult at present for other authors to duplicate these data given the proprietary homemade reagents

We appreciate the reviewer's comments, since the question of potential contaminants is often a concern in describing novel activities of a protein. Given that protein preps purified from cells are seldom (if ever) truly 100% pure, the critical question is whether there are contaminants that confound the results. In the present study, we are confident that the activities described herein are *bona fide* functions of CCN1 based on three broad lines of evidence using different analytical approaches:

1. Biochemical approach. Our protein preparations are evaluated using SDA-PAGE, which allowed us to estimate the proteins to be >90-95% pure. However, the possibility of minor impurities that are not clearly visible on a gel cannot be excluded by this criterion. We have also measured the level of endotoxin using the Limulus amoebocyte lysate (LAL) method and further purified the protein through a polymyxin-B-agarose column to remove potential endotoxins (see response to Reviewer 2 #4 below). Results from these experiments excluded potential endotoxin involvement. Another compelling argument

against the possible contribution of contaminants is the fact that multiple preparations of CCN1 expressed in very different cellular sources and purified using distinct methodologies all have the same activities. Thus, our own CCN1 preparation (expressed in Sf9 insect cells and purified through sequential ion-exchange chromatography) and those from Novus Biologicals (expressed in *E. coli* and purified by HPLC) and R&D Biosystems (expressed in CHO cells as a Fc-chimera and purified by protein A affinity chromatography) all bind bacterial patterns (PGN and LPS) and TLR2 and TLR4 similarly (see below and Supplementary Fig. S8). Furthermore, all these CCN1 protein preps induced phagocytosis of bacteria in a cilengitide-inhibitable manner (Supplementary Fig. S8). It is highly unlikely that proteins expressed in insect cells, *E. coli*, and hamster cells and purified using different chromatographic techniques will have the same confounding contaminants. Additionally, fragments of CCN1 representing specific domains prepared using different procedures also show binding to *S. aureus* (Fig. 1c) and bacterial pattern PGN (Supplementary Fig. S3).

In addition, mutant CCN1 proteins (CCN1-D125A and CCN1-DM) alter the opsonin functions of CCN1 as predicted. Thus, CCN1-D125A is defective for binding $\alpha\text{v}\beta 3$ and therefore unable to induce phagocytosis (Fig. 2b,f). CCN1-DM cannot bind LPS and is therefore unable to induce opsonophagocytosis of *P. aeruginosa* (Figs. 1e,g and 2f). CCN1-DM only binds TLR4 and not TLR2, and therefore induces only TLR4-dependent inflammatory gene expression that is unaffected by *Tlr2* knockout (Fig. 8d,f). These mutant protein studies reveal activities that are clearly attributed to the CCN1 polypeptide and not to unknown contaminants.

2. Genetic approach. An important line of evidence for the opsonin activity of CCN1 is based on mouse genetic models. Mice with *Ccn1* deletion in myeloid cells (*Ccn1* ^{Δ Myeloid}) are defective for bacterial clearance (Fig. 4), in accordance with the *in vitro* functions of CCN1 as an opsonin (Figs. 2,3). As predicted by our proposed mechanism, *Ccn1*^{D125A/D125A} knockin mice are also defective for clearance of *S. aureus*, whereas *Ccn1*^{DM/DM} mice are not (Fig. 5a,b). *Ccn1* ^{Δ Myeloid} mice display reduced inflammatory response (Fig. 6), consistent with the direct binding of CCN1 to TLR2 and TLR4 to induce inflammation in a MyD88-dependent manner (Figs. 7,8). Together, the genetic evidence is completely consistent with, and in support of, the opsonin and inflammation functions of CCN1 described herein.
3. Pharmacological and immunological approach. Antibody depletion of CCN1 using specific anti-CCN1 antibodies obliterated CCN1-induced inflammatory gene expression (Supplementary Fig. S14), and cilengitide completely blocked CCN1-induced phagocytosis (Fig. 2d and Supplementary Fig. S8). These data further show that these functions are attributed to the CCN1 polypeptide.

As mentioned above, we have tested the reproducibility of our results using CCN1 protein purchased from commercially available sources, as suggested by the reviewer. We tested CCN1 expressed in *E. coli* (Novus Biologicals, NBP2-34944) and in CHO cells (R&D Biosystems, 4055-CR-050). Both were able to bind LPS and PGN as well as TLR2 and TLR4, similar to our own CCN1 preparation (Supplementary Fig. S8). They also stimulated phagocytosis of *S. aureus* in a cilengitide-inhibitable manner (Supplementary Fig. S8). Thus, our findings can be replicated using commercially prepared CCN1 isolated from completely different cellular sources.

3. please provide catalog #s of inhibitors.

As suggested, we have incorporated the catalog numbers of inhibitors used in the Materials and Methods section of the manuscript.

4. siRNA studies, was only one siRNA used? It does not appear that westerns were used to assess knockdown

We have tested at least 2 independent siRNAs pre-designed by IDT (Integrated DNA Technologies) and they showed similar knockdown efficiency. Successful knockdown of individual integrin subunit was confirmed by qPCR analysis as shown in Supplementary Fig. S6.

5. can the authors specifically comment on the role of endogenous full length CCN1 versus exogenously added CCN1 in the activities reported here. Are the levels of exogenous CCN1 physiologically relevant?

As a matricellular protein, secreted CCN1 is rapidly associated with the ECM and only a small fraction is found in the cell media (Yang and Lau, *Cell Growth Differ.* 2:351-7, 1991). In most tissues, CCN1 is expressed at a low level in homeostasis but is dramatically increased in a local fashion at the site of injury upon damage, with contribution from platelet α -granule release and secretion from activated myeloid cells (Refs. 25, 73). Since CCN1 is highly inducible, locally expressed, and ECM-associated, it is difficult to ascertain its physiological bioactive concentration. Circulating levels of CCN1 in biofluids, where detected, may preferentially contain degraded protein fragments. However, the physiological relevance of CCN1 is clearly demonstrated by the defects in bacterial clearance observed in *Ccn1* ^{Δ Myeloid} and *Ccn1*^{D125A/D125A} mice (Figs. 4,5). Exogenous CCN1 administered systemically may be required at a higher level compared to locally expressed, ECM-associated endogenous CCN1.

6. Please comment on how to advance these ideas therapeutically eg. regarding CCN1 domains or peptides

The ability of CCN1 to promote bacterial clearance by opsonophagocytosis suggests potential therapeutic value for the treatment of antibiotic-resistant infections. Aside from application of the full-length protein, it may be possible to construct a simplified CCN1 polypeptide that includes the integrin α v β 3 binding site linked to the binding sites for LPS and/or PGN. Further studies will be needed to better define the minimal functional domains and to assess the efficacy of such therapeutic peptides. We have added this to the Discussion section (last paragraphs).

Reviewer #2 (Remarks to the Author):

This paper aims to show that CCN1 is an opsonin for Gram negative as well as Gram positive bacteria.

1. The authors claim that CCN1 bind to LPS, peptidoglycan, TLR2, TLR4 and as well to α v β 3. This would be a highly remarkable and therefore such statement should be made in the context of proper observations with solid binding studies. To my surprise the authors only show

immunoassay results. They need to show affinity for all of these ligands, generated by SPR, ITC or similar solid binding-analysis.

We thank the reviewer for the insightful comments and helpful suggestions. The idea that CCN1 has many functions and binding partners does indeed appear remarkable. In this context, it is important to recognize that CCN1 is a modular protein with 4 discrete domains with homologies to 4 distinct protein families, thereby bringing together multiple functional domains from diverse proteins in a compact arrangement. The matricellular nature of CCN1 also underscores its ability to interact with proteins and polysaccharides.

As suggested by the reviewer, we have successfully used SPR to analyze CCN1 interaction with LPS, TLR2, TLR4, and integrin $\alpha\beta 3$ and determined the binding affinities of each interaction, thus greatly strengthening the binding studies in the manuscript. We have conducted these SPR binding studies using CCN1 as both the ligand and the analyte, although the data presented in the manuscript employed CCN1 as the analyte (Figs. 1h, 8b,c, and Supplementary Fig. S13). This is because amine coupling of CCN1 to a standard CM5 sensor chip would destroy many of its lysine residues critical for function, whereas antibody capture would obfuscate the vWC domain against which our anti-CCN1 antibodies were raised. Interestingly, the results obtained are completely consistent with our expectations as summarized below.

For binding to LPS, we immobilized LPS on a Biacore HPA hydrophobic chip and used CCN1 as an analyte. The SPR result showed CCN1 specifically binds to LPS with a K_D value of 67.4 nM (Fig. 1h,i). However, PGN could not be readily analyzed using this method because intact PGN from Gram-positive bacteria is insoluble due to extensive cross-linking of the peptide side chains and exists in suspension in water. Very few SPR studies on PGN binding exist, and they are highly specialized studies in which PGN is solubilized by hydrolysis and/or enzymatic digestion (Asong et al, JBC 284:8643-8653, 2009). Such approaches significantly alter the structure of intact PGN and are beyond the scope of our current study. However, all our SPR results for CCN1 binding to LPS, TLR2, TLR4, and $\alpha\beta 3$ (see below) are completely consistent with results from our solid-phase binding assays, immunoblot assays, and functional assays. Thus, we are confident that our results for PGN binding accurately reflect CCN1 activity.

For protein-protein interactions, we immobilized TLR2, TLR4, and integrin $\alpha\beta 3$ as ligands on CM5 sensor chip, and used CCN1 as the analyte. These studies showed that CCN1 binds specifically to TLR2, TLR4, and $\alpha\beta 3$ with K_D values of 227 nM, 291 nM, and 202 nM, respectively, confirming direct binding of CCN1 to these receptors. These new results are added to Fig. 8b,c, and Supplementary Fig. S13. The binding affinity of CCN1 to TLR4 is slightly stronger than that of another DAMP, HMGB1, which has a K_D of 650 nM for binding TLR4 in the absence of MD2 based on SPR analysis (Ref 72).

As mentioned above, we have also performed SPR analysis with CCN1 as the ligand using two different methods of immobilization. First, CCN1 was immobilized on CM5 chips by amine coupling via lysine residues. Since mutations in a cluster of lysine residues in the CT domain in the CCN1-DM mutant abolished binding to LPS and TLR2, we anticipated that amine coupling would block CCN1 binding to TLR2. Indeed, CCN1 coupled to CM5 did not bind TLR2, as we expected (Fig. A below). It also did not bind to TLR4, suggesting that lysine residues are also important for interaction with TLR4. However, CCN1 in this configuration did show binding to integrin $\alpha\beta 3$, the binding site for which does not involve a lysine residue and is located in the vWC domain (Fig. A below). To avoid issues with lysine modification, we instead used an antibody capture protocol by coupling anti-CCN1 antibodies (raised against vWC domain of

CCN1) to the sensor chip to capture CCN1 without lysine modification. Using this configuration, we found specific binding of CCN1 to TLR2 and TLR4 (Fig. A). However, CCN1 failed to bind integrin $\alpha\beta_3$ in this construct as expected because the vWC domain is pre-engaged by the anti-CCN1 antibody. These results are completely consistent with our expectations, and reinforce our data using CCN1 as analyte. Taken together, CCN1 clearly binds TLR2, TLR4, and $\alpha\beta_3$ as demonstrated by SPR analyses. In the manuscript, we showed SPR data using CCN1 as the analyte to avoid complications related to CCN1 immobilization.

Figure A. CCN1 binds to TLR2, 4, and integrin $\alpha\beta_3$ in SPR analysis. CCN1 was either directly immobilized by amine coupling on CM5 sensor chip (*upper*) or indirectly captured by anti-CCN1 antibody that was immobilized (*lower*). CCN1 only bound integrin $\alpha\beta_3$ when immobilized on CM5 chip but did not bind TLR2 and TLR4 due to modification of critical lysines. By contrast, TLR2 and TLR4 bound to CCN1 captured through anti-CCN1 antibody, whereas integrin $\alpha\beta_3$ did not bind because the antibody targeted the vWC domain where the $\alpha\beta_3$ binding site is located.

2. Also for the purpose of estimating how much CCN1 binds they should compare binding of CCN1 on bacteria (flow cytometry) to established opsonins as antibodies and complement. How much is binding and how strong is it binding.

As suggested, we have used flow cytometry to compare the binding of CCN1 and the human C3b complement protein to heat-killed *S. aureus*. We observed dose-dependent binding of both CCN1 and C3b to *S. aureus* (Supplementary Fig. S2), and CCN1 appears to bind more efficiently than C3b in this assay. Over 70% of *S. aureus* was bound at 5 $\mu\text{g/ml}$ CCN1, whereas ~27% was bound by 5 $\mu\text{g/ml}$ C3b. Thus, CCN1 binding to *S. aureus* is clearly demonstrated by solid-phase binding (Fig. 1a-c) and flow cytometry (Supplementary Fig. S2). However, the relative binding strengths between CCN1 and C3b cannot be assessed quantitatively as factors such as the affinities of the antibodies used (polyclonal anti-CCN1 vs. monoclonal anti-iC3b) are important variables.

3. Proper opsonins only bind to cells when bound to bacteria (as antibodies and complement do) Otherwise this protein would always activate TLR2 and 4. Authors should show

What is the mechanism of this, and do provide data to show this. Same hold true for interaction with $\alpha\beta3$.

CCN1 is expressed at a low level in most tissues during homeostasis and becomes highly induced upon injury, infection, and inflammation. This expression pattern is very different from that of other opsonins such as antibodies and complements, which are present at high concentrations even in homeostasis and therefore their binding to cells must be regulated. By contrast, CCN1 is highly regulated at the level of synthesis and secretion, and its binding to $\alpha\beta3$ and TLRs will only occur acutely upon infections and injuries.

SPR analyses showed that CCN1 has a higher affinity for LPS than for $\alpha\beta3$ or TLR2 and TLR4, and therefore it may bind bacteria before it binds cell surface receptors. This observation is in line with the reviewer's point. However, CCN1 affinities for $\alpha\beta3$ and TLR2/4 appear similar based on SPR data. Therefore, its binding to these receptors may be concurrent during injury and infection.

CCN1 binding to integrin $\alpha\beta3$ can occur even without the presence of bacteria or phagocytic cargos. We have previously shown CCN1 alone added to cells could induce p130^{Cas}-CrkII complex formation through integrin $\alpha\beta3$, leading to Rac1 activation (Ref 25). CCN1 also acts as a bridging molecule to induce phagocytosis of apoptotic cells, which requires CCN1 to bind $\alpha\beta3$ without the presence of a bacterial cargo.

4. Please provide data on purity of protein, including proper endotoxin measurements. I am not convinced by what is reported.

Since our CCN1 protein was expressed in Sf9 insect cells, the possibility of bacterial endotoxin contamination is unlikely. However, many protein preparations may contain traces of endotoxin from water, serum, plasticware, and other cell culture reagents. We have tested all CCN1 preps (including CCN1 mutants) for endotoxin using the Limulus amoebocyte lysate (LAL) method. We found that the endotoxin level is <0.08 EU/ μg protein (Supplementary Fig. S14), which is a trace amount typical of non-bacterial protein preparations. Furthermore, our CCN1 protein preps retained the ability to induce inflammatory gene expression after passage through a polymyxin B-agarose column, indicating that the observed effect is not due to endotoxin contamination (Supplementary Fig. S14). Please also see discussion on potential contaminants above (responses to Reviewer 1 #2).

5. Although most of the mice data are solid, this paper needs mechanistic experiments to make the claims that the authors try to make.

We appreciate the recognition that our mice data are solid. We have laid out the mechanism of CCN1 action in opsonization and inflammation in the manuscript, as summarized below and diagrammatically depicted in Fig. 9. Each of the mechanistic steps described below is supported by experimental evidence.

1. The mechanism by which CCN1 acts as an opsonin is based on its direct binding to bacteria and to activate phagocytosis. CCN1 binds LPS through its CT domain and PGN through its vWC and TSR domains, thereby opsonizing Gram-negative and Gram-positive bacteria, respectively (Fig. 1). CCN1 also induces phagocytosis by binding integrin $\alpha\beta3$ to activate Rac1, thus leading to engulfment of the opsonized bacteria.

Supporting this mechanism is the fact that a single amino acid substitution in CCN1 disrupting binding to $\alpha\beta3$ is defective for the induction of phagocytosis (Fig. 2), and knockin mice expressing this mutant suffer severe defects in bacterial clearance (Fig. 5). The mechanism by which CCN1 activates Rac1 has been previously described (Ref 25), and is mediated through CCN1 binding to integrin $\alpha\beta3$, triggering the formation of the integrin-p130^{Cas}/CrkII complex and recruitment of DOCK180, a Rac1-activating guanine nucleotide exchange factor (GEF).

2. After phagocytosis, CCN1 promotes the killing of engulfed bacteria by inducing bactericidal ROS production. Mechanistically, this occurs through CCN1 activation of Rac1 as outlined above. Rac1, in turn, activates NOX2 to produce superoxide (Fig. 3).
3. The mechanism by which CCN1 acts in bacterial clearance *in vivo* is supported by data in animal models. Deletion of *Ccn1* in myeloid cells leads to defects in bacterial clearance (Fig. 4), indicating the importance of *Ccn1* expressed in macrophages and neutrophils in its mode of action. Further, *Ccn1*^{D125A/D125A} knockin mice also suffer similar defects in bacterial clearance (Fig. 5). This shows that CCN1 activities in bacterial clearance *in vivo* is mediated through $\alpha\beta3$ -dependent activities, which include opsonophagocytosis and ROS generation as described herein.
4. The mechanism by which CCN1 induces sterile inflammation is through its direct binding and activation of TLR2 and TLR4, as shown by solid-phase binding assay, immunoblot assay, and SPR analysis (Fig. 8). Binding of CCN1 to TLR2 and TLR4 induces MyD88-dependent inflammatory responses *in vivo* and *in vitro* (Figs. 7,8). That CCN1 induces inflammatory responses through TLR2 and TLR4 is shown in *Tlr2*^{-/-} and *Tlr4*^{-/-} BMDMs, as well as in *Myd88*^{-/-} BMDMs (Fig. 8).

Reviewer #3 (Remarks to the Author):

Jun and Lau investigate the ability of CCN1, a matricellular protein known to be involved in a number of key host functions, to function as a host opsonin and elicit components of the inflammatory response. The authors ultimately use mouse models to demonstrate that CCN1 has an important role in the host innate immune response. Overall, the work is interesting and a significant advance for the field. I have a few comments for the authors to consider.

1. It would be optimal to measure production of extracellular superoxide by reduction of cytochrome c, a traditional quantitative assay (see Bylund et al., *Methods Mol Biol.* 2014;1124:321-38). This would provide information about the quantity of superoxide produced by macrophages (e.g., as in Figure 3). This assay is superior to the DHE assay for quantitation and measures extracellular superoxide—which I presume is the process for cells activated by CCN1.

We thank the reviewer for the valuable comments. After phagocytosis of bacteria in macrophages, CCN1 treatment increases ROS production and ROS-dependent bacterial killing. CCN1 engagement of $\alpha\beta3$ activates Rac1, which in turn activates Nox2. Nox2 is located on the plasma membrane and can therefore generate superoxide in the extracellular space. During phagocytosis, part of the membrane undergoes invagination to form the phagosome, thereby bringing some of the Nox2 enzyme complexes into the phagosome where Nox2 can generate superoxide inside the phagosome. Therefore, CCN1-induced ROS can be both intracellular and extracellular. In addition to measuring the CCN1-induced ROS using the DHE assay (Fig. 3), we

have also measured the CCN1-induced extracellular superoxide using the cytochrome c reduction method as suggested by the reviewer (Fig. B). As expected, CCN1-WT induces extracellular superoxide production, whereas CCN1-D125A is defective since it is unable to bind $\alpha\beta3$ or activate Rac1.

Figure B. The production of extracellular superoxide was measured in BMDMs treated with either CCN1-WT or CCN1-D125A using the cytochrome c reduction method.

2. Protection related to promoting phagocytosis of *S. aureus* in the mouse model is not likely to be as significant in humans. This is because laboratory mice typically lack *S. aureus*-specific antibody, whereas humans contain antibody specific for *S. aureus*. Most notably, *S. aureus* is ingested readily by human phagocytes (neutrophils) in whole blood or in vitro following opsonization with normal human serum. Thus, *S. aureus* vaccine approaches that are directed to promote opsonophagocytosis often work well in mouse models, but fail in human systems, including past clinical trials. The ability of CCN1 to activate TLR2 and TLR4 is perhaps more important in context of a potential therapeutic for *S. aureus* infections. These points should be considered by the authors—probably in the discussion section.

We appreciate the reviewer's comments. It is true that vaccines against *S. aureus* have been ineffective in humans, partly because most humans already have antibodies against *S. aureus*. In addition, *S. aureus* also produces protein A on the cell surface, which captures the Fc of antibodies and blocks Fc receptor-mediated phagocytosis. For this reason, antibody-independent opsonization, such as CCN1-mediated opsonophagocytosis, may be more effective in humans. We have added this aspect to the Discussion (last paragraph).

3. It would be optimal to test/verify the ability of CCN1 to function with human phagocytic cells in vitro, and/or elicit inflammatory responses with human cells in vitro.

As suggested by the reviewer, we have assessed CCN1-induced phagocytosis and inflammatory gene expression in differentiated THP.1 human macrophage. These results have been incorporated into Fig. 2e and Supplementary Fig. S12.

REVIEWERS' COMMENTS:

Reviewer #1 (Remarks to the Author):

The revisions are acceptable

Reviewer #2 (Remarks to the Author):

The authors have performed most if not all of my suggested experiments, data are convincing. However the overall conclusion is now no longer supported by their data
The conclusion -from the authors themselves- is that CNN1 is perfectly capable of interacting with TLR's and activate cells and ALSO can bind bugs. Nowhere in the manuscript it is shown that the DUAL interaction is required. In fact, in answer to my question they argue that CNN1 is stringently regulated and that it then can interact with the receptors

Therefore: CNN1 is NOT an opsonin. It is a cell activator, nice story also, but therefore probably useless as a therapeutic

Authors should reframe their conclusion based upon their own, very good, data

Reviewer #3 (Remarks to the Author):

Thank you for your responses to my comments and suggestions.

Responses to Reviewers' comments:

Reviewer #1 (Remarks to the Author):

The revisions are acceptable

We thank the reviewer for the favorable assessment of our revised manuscript.

Reviewer #2 (Remarks to the Author):

The authors have performed most if not all of my suggested experiments, data are convincing.

We thank the reviewer for his helpful suggestions in the previous review and the recognition that our data are convincing.

However the overall conclusion is now no longer supported by their data

The experiments suggested by the reviewer were designed to verify our conclusions. Specifically, the reviewer suggested that we conduct SPR analysis to further establish CCN1 binding to TLRs, reinforcing the solid-phase binding and immunoblot assays for receptor binding. In addition, flow cytometry was done to show that CCN1 can bind bacteria in solution. The data from these additional experiments are convincing and fortify our findings using other techniques, thereby greatly strengthening our overall conclusions.

The conclusion -from the authors themselves- is that CNN1 is perfectly capable of interacting with TLR's and activate cells and ALSO can bind bugs. Nowhere in the manuscript it is shown that the DUAL interaction is required. In fact, in answer to my question they argue that CNN1 is stringently regulated and that it then can interact with the receptors

Indeed, we have shown that CCN1 is perfectly capable of binding TLRs and activating TLR signaling (Figs. 7,8). These findings support the conclusion that CCN1 can function as a DAMP and activate inflammatory responses through TLRs. However, the ability of CCN1 to activate TLRs in no way invalidates its ability to induce bacterial clearance by opsonization, which is a separate and distinct activity.

Our manuscript is organized into two parts that describe two major conclusions: 1. CCN1 functions as an opsonin for bacterial clearance (Figs. 1-5), and 2. CCN1 binds and activates TLRs (Figs. 6-8). These two activities are mediated through distinct receptors: $\alpha\text{v}\beta\text{3}$ mediates phagocytosis and bacterial killing, whereas TLR2/4 mediate inflammatory response. These two activities of CCN1 can function independently, and one activity does not depend on the other, even though both functions can occur upon injury when CCN1 expression is induced. For example, the CCN1-D125A mutant that is defective for inducing phagocytosis is fully capable of binding and activating TLRs (Fig. 8d,e, Supplementary Figs. 10,11). We have not suggested that the opsonin and TLR activation functions of CCN1 are interdependent in any way.

To avoid conflation of these two distinct activities, we have stated in the last paragraph of the Introduction: "Independent of its opsonin activity, CCN1 can also directly bind and activate TLR2 and TLR4 to induce the expression of proinflammatory cytokines in a MyD88-dependent manner." CCN1 is both an opsonin and an activator of TLR signaling; the fact that CCN1 has multiple activities does not negate its function as an opsonin.

Therefore: CCN1 is NOT an opsonin. It is a cell activator, nice story also, but therefore probably useless as a therapeutic. Authors should reframe their conclusion based upon their own, very good, data.

The Merriam-Webster Dictionary defines an opsonin as “any of various proteins (such as antibodies or complement) that bind to foreign particles and cells (such as bacteria) making them more susceptible to the action of phagocytes.” The abilities of CCN1 to bind bacteria and induce bacterial clearance through $\alpha\beta3$ -mediated phagocytosis and ROS generation have been extensively documented in this manuscript (Figs. 1-5). Therefore, CCN1 fits the dictionary definition of an opsonin perfectly, and there is no better term to describe its functions in bacterial clearance.

To reiterate the ample evidence of the CCN1 as an opsonin:

1. CCN1 binds *S. aureus* and *P. aeruginosa* through specific binding of PAMPs of Gram-positive (PGN) and Gram-negative (LPS) bacteria (Fig. 1, Supplementary Fig. 1-4).
2. CCN1 induces the phagocytic clearance of *S. aureus* and *P. aeruginosa* through the engagement of integrin $\alpha\beta3$ (Fig. 2, Supplementary Fig. 5-8).
3. CCN1 stimulates the production of bactericidal ROS, promoting bacterial killing after phagocytosis through $\alpha\beta3$ -dependent Rac1-NOX2 activation (Fig. 3).
4. Myeloid deletion of *Ccn1* in mice leads to increased susceptibility to bacterial infections, resulting in elevated mortality and organ colonization (Fig. 4, Supplementary Fig. 8-9).
5. Knock-in mice expressing CCN1-D125A unable to bind $\alpha\beta3$ experience similar defects in bacterial clearance, indicating the crucial role of $\alpha\beta3$ -dependent CCN1 functions – phagocytosis and ROS generation – in bacterial clearance *in vivo* (Fig. 5).

The multifunctional nature of CCN1 is characteristic of matricellular proteins and has been extensively demonstrated in previous studies (see CCN1 reviews, refs. 29,30). In addition to antibodies and complement proteins, several other families of proteins, including collectins, ficolins, and pentraxins can also function as opsonins and have been recognized as such. Many opsonins are also multifunctional. The fact that CCN1 has multiple activities, including activation of TLRs, does not nullify its function as an opsonin.

The therapeutic potential of CCN1 is evidenced by the observation that the administration of CCN1 protein effectively promotes bacterial clearance in infected mice (Fig. 5d,e). Both Reviewers 1 and 3 have pointed to the potential therapeutic value of CCN1, which we have discussed based on their suggestions. Future clinical studies will be required to better assess the therapeutic potential of CCN1.

Reviewer #3 (Remarks to the Author):

Thank you for your responses to my comments and suggestions.

We thank the reviewer for the favorable assessment of our revised manuscript.